# Utilizing *C. elegans* Spermatogenesis and Fertilization Mutants as a Model for Human Disease

**DOI:** 10.3390/jdb13010004

**Published:** 2025-01-25

**Authors:** Sofia M. Perez, Helena S. Augustineli, Matthew R. Marcello

**Affiliations:** Biology Department, Pace University, New York, NY 10038, USA

**Keywords:** *Caenorhabditis elegans*, spermatogenesis, fertilization, genetics, disease modeling, phenotype, ortholog

## Abstract

The nematode *C. elegans* is a proven model for identifying genes involved in human disease, and the study of *C. elegans* reproduction, specifically spermatogenesis and fertilization, has led to significant contributions to our understanding of cellular function. Approximately 70 genes have been identified in *C. elegans* that control spermatogenesis and fertilization (*spe* and *fer* mutants). This review focuses on eight genes that have human orthologs with known pathogenic phenotypes. Using *C. elegans* to study these genes has led to critical developments in our understanding of protein domain function and human disease, including understanding the role of *OTOF* (the ortholog of *C. elegans fer-1*) in hearing loss, the contribution of the *spe-39* ortholog *VIPAS39* in vacuolar protein sorting, and the overlapping functions of *spe-26* and *KLHL10* in spermatogenesis. We discuss the cellular function of both the *C. elegans* genes and their human orthologs and the impact that *C. elegans* mutants and human variants have on cellular function and physiology. Utilizing *C. elegans* to understand the function of the genes reviewed here, and additional understudied and undiscovered genes, represents a unique opportunity to understand the function of variants that could lead to better disease diagnosis and clinical decision making.

## 1. Introduction

### 1.1. C. elegans as a Model Organism for Human Disease

The roundworm *Caenorhabditis elegans* has been used extensively to understand the impact of human genetic variation on protein function and is a proven model for identifying genes involved in disease pathogenesis [1,2,3,4]. *C. elegans* is a valuable model organism for a multitude of reasons, including the ability to (1) easily manipulate its genome for forward genetic screening, transgenic animal construction, and mutation mapping, (2) observe live cellular and molecular events using fluorescently-tagged proteins in a transparent organism, (3) perform genome-wide RNA interference (RNAi) screens, (4) conduct screens for suppressor and enhancer modifiers of genes that cause disease, and (5) employ high-throughput analysis, including the use of microfluidic platforms [5,6]. In addition, CRISPR/Cas9 gene editing is used abundantly in *C. elegans* to generate precision-modified alleles, which can be used to investigate disease etiology and treatment [6]. Many cellular processes and molecular pathways, particularly cell signaling, are well conserved between *C. elegans* and humans, which allows for mechanistic discoveries and direct functional studies [6,7]. The use of model organisms to understand human phenotypes has led to significant advances of our understanding of human disease and its underlying biology [8]. The use of *C. elegans* as a model system has contributed significantly to our understanding of human disease including neurodegenerative/neuromuscular disorders, metabolic disorders, kidney disease, and cancer [4]. Overall, *C. elegans* is a popular model organism for the study of human disease that can be used in innovative and creative ways to accelerate personalized and precision medicine [1].

### 1.2. C. elegans Spermatogenesis and Fertilization as a Model for Cellular Function

The *C. elegans* germline and reproductive tract is an excellent model for a variety of cell biological processes [9,10]. The study of *C. elegans* reproduction is particularly attractive because of the short reproductive cycle (~3.5 days), large number of progeny produced, and the ability to observe oocyte development, ovulation, fertilization, and early embryogenesis in intact animals [11]. Moreover, greater than 20% of all known *C. elegans* genes are expressed in the hermaphrodite germline [11]. *C. elegans* have two sexes: hermaphrodites and males [12]. Hermaphrodites produce 300–400 sperm during their final larval stage before switching to oocyte production [13]. Individual males produce sperm continually, and single males can sire over one thousand progeny [13,14]. *C. elegans* reproduction can occur via self-fertilization in hermaphrodites or by mating with males to produce outcross progeny [15]. Both sexes produce round spermatids, which must undergo a maturation process named sperm activation or spermiogenesis (Figure 1) [16]. Sperm activation results in mature sperm that are motile and capable of fusion with the oocyte [16]. One of the hallmark processes of sperm activation is the fusion of the Golgi-derived membranous organelles (MOs) with the plasma membrane, which allows for both the redistribution of proteins necessary for fertilization and the secretion of glycoproteins whose functions remain largely uncharacterized [16,17,18]. MOs are specialized secretory vesicles within spermatids that fuse with the spermatid plasmid membrane during spermiogenesis [19]. MO fusion is similar to the acrosome reaction in the sperm of many mammalian species [20].

This review will focus on a class of *C. elegans* mutants with spermatogenesis (*spe* mutants) and fertilization (*fer* mutants) defects (Table 1). *spe* and *fer* mutants produce spermatocytes, spermatids, and/or sperm that cannot function properly during spermatogenesis, spermiogenesis, and/or fertilization [24]. *C. elegans* mutants isolated because their sperm were unable to fertilize an egg were originally named *fer* mutants [25]. However, this terminology is obsolete, and all new mutants that have functionally defective sperm are called *spe* mutants [21,26]. In *spe* mutants, the sperm function defect in these animals results in infertility, with only a few progenies produced [26]. In *spe* hermaphrodites, most oocytes remain unfertilized due to their defective sperm. However, since their oocytes are unaffected, *spe* hermaphrodites can produce viable progeny when crossed with wild-type males [26].

### 1.3. Human Orthologs of C. elegans fer and spe Genes

The identification of orthologs is essential for comparative studies to interpret and infer conclusions from model organism studies [27]. For the purposes of this review, we have focused on the common set of orthologs in The Alliance of Genome Resources (the Alliance) database (www.alliancegenome.org, 5 December 2023) (Table 1). The ortholog inferences from the Alliance are included from methods benchmarked by the Quest for Orthologs Consortium and manually curated inferences from HGNC and ZFIN [27]. There are many ortholog inference methods, with new methods in development [28]. The ortholog inferences are not complete and there will be omissions, including orthologs identified from iterative HMMer approaches [29]. However, the Alliance database has harmonized cross-organism data that are consistent and include data from seven knowledgebase projects including WormBase [29]. The human orthologs of *C. elegans fer* and *spe* genes were retrieved from the Alliance of Genome Resources (https://www.alliancegenome.org; 5 December 2023, Version 6.6.0), using the highest “Stringency” filter [Stringent (default): Primary criterion: An ortholog called by three or more methods is included if it is a best count OR a best reverse count. Secondary criteria: An ortholog predicted by ZFIN, HGNC, or Xenbase is always included, regardless of count. An ortholog called by two methods is included if it is both a best count AND a best reverse count] [29]. This review is focused on human orthologs with gene–phenotype relationships in Online Mendelian Inheritance in Man (OMIM); however, there are additional orthologs of *C. elegans* genes that do not have associated human disease phenotypes, including orthologs of *fer-1*, *mib-1*, and *uba-1* (Table 1). OMIM is a database of primary data that employs expert review of the biomedical literature to synthesize and summarize new and important information [30]. Allelic variants of disease-related genes are included in OMIM; however, only selected mutations are included [30]. As a result, the variants discussed in this review are not comprehensive and there are other sources for identifying variants associated with disease genes, including the Alliance and ConVarT [29,31].

Valuable information can be gained about individual protein domain function by studying the variants of proteins [4,6,32,33,34]. We have focused on how the sequence variants in both *C. elegans* and human genes can impact protein function, to outline how their analysis could lead to a better understanding of how these proteins function. The progress in next-generation sequencing technologies has resulted in a growing number of human genetic variants [31]. Unfortunately, the functional effects of those variants remain unknown [31]. Understanding these genetic variants is necessary for disease diagnosis and therapeutic decisions, and model organisms, such as *C. elegans*, can function as valuable models to understand these variants [31].

## 2. *C. elegans fer* and *spe* Genes and Human Ortholog Function

### 2.1. C. elegans FER-1 and H. sapiens DYSF, MYOF, and OTOF

#### 2.1.1. Ferlin Family Protein Function

The proteins encoded by *fer-1* and its human orthologs *dysferlin* (*DYSF*), *myoferlin* (*MYOF*), and *otoferlin* (*OTOF*) are members of the ferlin family of proteins (Figure 2). *C. elegans fer-1* is the founding member of the ferlin gene family [35]. The ferlin family of proteins are a group of intrinsic membrane proteins necessary for vesicle fusion [36]. The protein domains included in ferlin family members vary and there is a lack of consensus regarding domain boundaries [37,38]. The ferlins discussed here contain multiple C2 domains, a Ferlin A-domain (FerA), a Ferlin B-domain (FerB), and a Ferlin C-terminal domain (Figure 2). Ferlin family members are divided into two subgroups by their presence of a dysferlin (DysF) domain [37]. Type 1 ferlins include two DysF domains, where one DysF domain is inserted/embedded within a second DysF domain [37,39]. These nested DysF repeat sequences are divided into two parts, the dysferlin N-terminal region (DysF-N) and the dysferlin C-terminal region (DysF-C) [40]. Type 2 ferlins do not contain a DysF domain [37].

Members of the ferlin family generally contain several C2 domains, and multiple tandem C2 domains are the defining feature of the ferlin proteins [36,38,40]. C2 domains are found in a diverse set of eukaryotic proteins and were discovered in classical protein kinase C (PKC) as being responsible for calcium-dependent membrane binding [41]. The C2 domains bind phospholipids in membranes in both calcium-dependent and -independent manners [42,43,44,45]. The C2 domains in FER-1 do not contain the five conserved calcium-binding aspartate residues in the PKC and synaptogamin I C2 domains [36]. There is conflicting evidence as to whether these residues are necessary to coordinate calcium in the OTOF C2A domain [36,46,47].

FerA domains are typically defined as a 66 amino acid FerA consensus sequence composed of two α-helical segments [48]. Using that definition, OTOF does not contain a FerA domain, and as a result, this domain is not included in the protein domain diagrams [37]. Harsini et al. showed that if an additional primary sequence is included before and after the defined regions, then two additional consensus α-helices can be predicted in all ferlin proteins in the family [48]. The resulting predicted four-helix FerA domain is capped by disordered consensus residues between C2C on the amino-terminal side and the predicted FerB sequences on its C-terminal side [48]. As a result, the updated description for FerA has “four long amphipathic α-helices, with two groups of two α-helices separated by a long, central connecting linker” [48]. Using this updated FerA domain definition, the FerA domains of DYSF, MYOF, and OTOF have calcium-dependent, phospholipid-binding activity and can possibly interact with the membrane [48]. There are no data to indicate if the FerA domain of FER-1 has this activity.

**Figure 2 jdb-13-00004-f002:**
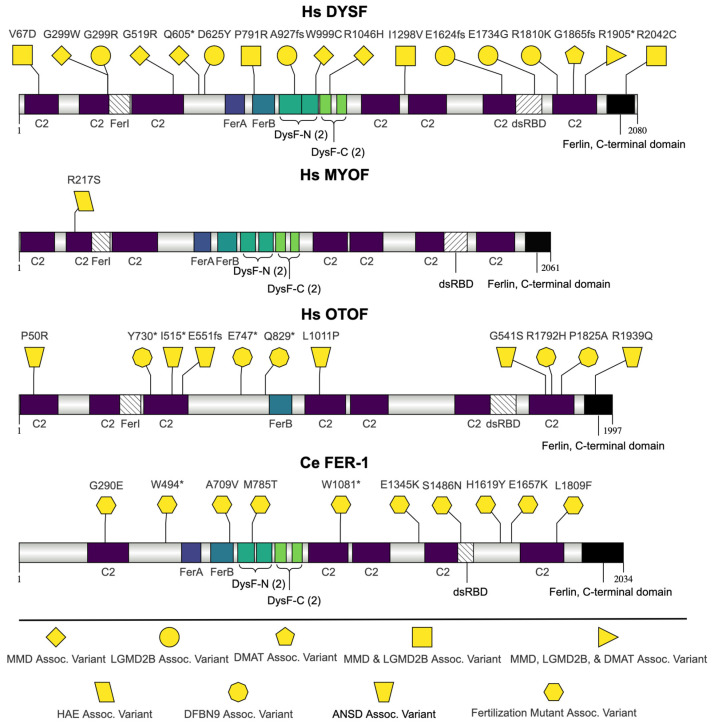
Protein diagram of *C. elegans* FER-1 and *H. sapiens* DYSF, MYOF, and OTOF. Diagram depicting full-length protein with protein domains and genetic variants. Human and *C. elegans* genetic variants are labeled with the shapes indicated at the bottom of the figure (asterisk (*) indicates a nonsense mutation). The FerI and dsRBD domains overlap with C2 domains, as designated by the diagonal lines in the domain depiction. The amino acid numbers correspond to isoform a. In DYSF, the G299W, G519R, Q605*, W999C, and R1046H mutations are associated with MMD; the G299R, D625Y, A927fs, E1624fs, E1734G, and R1810K mutations are associated with LGMD2B; the G1865fs mutation is associated with DMAT; the V67D, P791R, I1298V, and R2042C mutations are associated with MMD and LGMD2B; the R1905* mutation is associated with MMD, LGMD2B, and DMAT. In OTOF, the Y730*, E747*, Q829*, R1792H, and P1825A mutations are associated with DFBN9, and the P50R, I515T, G541S, E551fs, L1011P, and R1939Q mutations are associated with ANSD. Schematic diagram was generated using Illustrator for Biological Sequences (IBS) 2.0 [49].

The ferlin proteins discussed also contain a FerI domain and a double-stranded RNA-binding domain (dsRBD) that overlaps with a C2 domains. There is no defined role for the FerI domain, but it has been identified as a nucleolar protein domain [37,50]. The FerI domain for DYSF, MYOF, and OTOF is between the second and third C2 domains and overlaps with the second C2 domain [51]. FER-1 has previously been predicted to have a FerI domain, but this domain is no longer annotated in InterProScan [37,51]. The dsRNA-binding domain begins within the fifth C2 domain of DYSF, MYOF, and OTOF and the fourth C2 domain of FER-1 [51]. The role for the ds-RNA binding domain is currently unclear [38].

The functions of the remaining domains in the ferlin proteins are also not well-defined. Despite being used to define the classes of ferlin subgroups, the precise role of the DysF domain is unknown [52]. FER-1, DYSF, and MYOF are Type 1 ferlins containing a DysF domain, while OTOF is a Type 2 ferlin [37]. In the methylotrophic yeast *Pichia pastoris*, the DysF domains of Pex30p and Pex31p are essential for a normal peroxisome number and size [53]. However, in contrast to the ferlin members discussed here, the *P. pastoris* DysF domains are not embedded within another DysF domain [37]. The Ferlin C-terminal domain is found in proteins in the ferlin family and contains a transmembrane domain but has no other described function [51]. No function or interaction has been attributed to the FerB domains [37].

#### 2.1.2. FER-1 Function in *C. elegans*

*fer-1* hermaphrodite mutants make normal numbers of sperm and oocytes but no fertilization occurs [25]. *fer-1* sperm have abnormally short pseudopods and are non-motile [54]. Notably, their MOs fail to fuse with the plasma membrane [25,54,55]. Multiple isoforms of *fer-1* are expressed in the sperm (specifically primary spermatocytes), and FER-1 localizes to the MO of spermatids [40,56]. *fer-1* mutants are hypersensitive to intracellular calcium depletion, and calcium is necessary for MO fusion [40]. *fer-1* is also expressed in muscle, and mutants also have defects in muscle cholinergic synapse function similar to *DYSF* knockout mice [57,58]. Of the ten FER-1 mutations highlighted in Figure 2, four (p.Trp494Ter, p.Trp1081Ter, p.Glu1345Lys, and p.Glu1657Lys) are nonconditional; sperm with these genetic lesions never fuse their MOs [40]. Two of the non-conditional alleles (p.Trp494Ter and p. Trp1081Ter) are nonsense mutations that produce no FER-1 [40]. The remaining six mutations (p.Gly290Glu, p.Ala709Val, p.Met785Thr, p.Ser1486Asn, p.His1619Tyr, and p.Leu1809Phe) are temperature-sensitive mutations that fuse few to no MOs at the restrictive temperature [40].

#### 2.1.3. *fer-1* Orthologs Associated with Human Disease

##### DYSF Mutations: (a) Miyoshi Muscular Dystrophy (MMD), (b) Muscular Dystrophy, Limb-Girdle, Autosomal Recessive 2 (LGMD2B), and (c) Myopathy, Distal, with Anterior Tibial Onset (DMAT)

Dysferlinopathies include a spectrum of muscle diseases characterized by two major phenotypes, Miyoshi muscular dystrophy (MMD) and limb-girdle muscular dystrophy type 2B (LGMD2B), and two minor phenotypes, distal myopathy with anterior tibial onset (DMAT) and asymptomatic hyperCKemia [59]. In OMIM, asymptomatic hyperCKemia is not associated with *DYSF* mutations and will not be discussed here [30]. Mutations in *DYSF* can cause either the presentation of MMD or LGMD2B; however, clinical presentations can manifest in different muscles [60]. Patients with either disease show a slow progression of muscle weakness and elevation of serum creatine kinase concentration [59]. The mean age of onset for MMD is ~22 years, while LGMD2B occurs at ~28 years old [59]. The major difference between diseases is the muscles affected; MMD affects the upper and lower limbs, particularly the calf muscles, while in LGMD2B, weakness and atrophy occur in the pelvic and shoulder girdle muscles [59]. DMAT is characterized by the early onset (mean age 20 years) of distal muscle weakness in the muscles of the anterior compartment of the legs, causing foot drop [59]. There is a high degree of variability of disease in patients with dysferlinopathy, although the clinical presentations can be similar [61].

There are a relatively high number of allelic variants for *DYSF* that are listed in OMIM [30]. Of the 22 selected variants, 5 do not result in single amino acid changes in the DYSF protein and will not be discussed here. Many variants only have a clinical description of patients with the mutation, and the impact of the mutation on protein function has not been determined experimentally. Overall, the type of mutation or level of protein expression does not account for the variability seen in patients with dysferlinopathy [61].

The mutations associated with MMD are p.Gly299Trp, p.Gly519Arg, p.Gln605Ter, p.Trp999Cys, and p.Arg1046His. The p.Gly299Trp mutation is in the second C2 domain of DYSF and is hypothesized to interfere with folding [62]. The mutation results in protein aggregation in myotubules and amyloid deposits in the muscles [63,64]. The p.Gly519Arg mutation is the result of a transition mutation that activates an exonic cryptic acceptor site and alters splicing [65,66]. The p.Gln605Ter is a nonsense mutation resulting in a truncated protein [67]. The p.Trp999Cys variant is the most frequent dysferlinopathy variant [68]. Expression of the mutated protein varies. It is expressed at low levels in fibroblasts and displays variable expression in muscle, with some patients having no expression and others having cytoplasmic accumulations of mutated protein [69,70]. More recent studies have shown that patients with this mutation lack DYSF expression [68,71]. In transfected myoblasts, both the p.Trp999Cys and wild-type proteins colocalize with the integrin-linked-kinase binding protein, affixin [69]. In these experiments, p.Trp999Cys DYSF is expressed at the plasma membrane, and the mutant can interact with affixin [69]. The p.Arg1046His mutation also colocalizes with affixin in myoblasts, but its expression is in cytoplasmic speckles, not the plasma membrane [69]. Affixin expression is reduced in the sarcolemma of MMD and LGMD2B patients [69].

The mutations associated with LGMD2B are p.Gly299Arg, p.Asp625Tyr, p.Ala927fs, p.Glu1624fs, p.Glu1734Gly, and p.Arg1810Lys. In contrast to the p.Gly299Trp variant associated with MMD, the p.Gly299Arg variant is associated with LGMD2B [64]. The p.Gly299Arg mutation leads to protein aggregation, specifically in amyloid deposits in the muscle, by hypothetically affecting folding [62,63,64]. Other than the description of the amino acid change, the impact of the p.Asp625Tyr and p.Glu1734Gly variants has not been studied [72]. The p.Ala927fs variant is predicted to result in premature termination 21 amino acids downstream of the mutation [73]. In patients with this mutation, there is no DYSF expression in the sarcolemma of muscle fibers [74]. Heterozygous individuals have reduced protein levels but no clinical symptoms [74]. The impact of the p.Glu1624fs variant is predicted to cause a premature stop codon nine amino acids downstream but has not been studied further [35,75]. The molecular impact of p.Arg1810Lys (also known as p.Arg1831Lys) has not been determined experimentally. This variant could result in either disrupted protein function or altered splicing [76].

The mutations associated with both MMD and LGMD2B are p.Val67Asp, p.Pro791Arg, p.Ile1298Val, and p.Arg2042Cys. p.Val67Asp is predicted to cause major disruptions in protein structure [77]. Muscle sections from patients with p.Pro791Arg mutations have reduced amounts of DYSF protein [78]. Proline 791 is conserved in *C. elegans* FER-1 [78]. Muscle from patients with p.Ile1298Val mutations also have reduced amounts of DYSF [79]. Patients with p.Arg2042Cys mutations have weak DYSF expression [80]. Zebrafish embryos expressing p.Arg2042Cys mutations are unable to repair lesions in the sarcolemma because they cannot recruit DYSF and phosphatidylserine to the repair patch, leading to the hypothesis that the human phenotype could arise due to a failure to repair damaged membranes [81].

There are only two mutations associated with the DMAT phenotype alone or in combination with the MMD and LDMD2B phenotype. The p.Gly1865fs mutation is the only one associated with DMAT, and there is no description of the molecular effects of the mutation [67,82,83,84]. The only mutation associated with MMD, LGMD2B, and DMAT is p.Arg1905Ter, and there is no further description of the impact of the variant other than that it will lead to the premature termination of translation [85].

##### MYOF Mutations: Angioedema Hereditary, 7 (HAE) (Provisional)

Angioedema (AE) is characterized by episodes of cutaneous or submucosal edema, often in the skin, abdomen, and upper respiratory tract [86]. Hereditary angioedema (HAE) with mutations in the myoferlin gene (HAE-MYOF) is one of eight types of HAE [86]. MYOF is expressed in the endothelial cells lining intact blood vessels and can regulate VEGF [87]. The only mutation in MYOF highlighted is the p.Arg217Ser mutation in the C2B domain (Figure 2). In transfection assays, the mutation increased expression of VEGFR-2 on the plasma membrane [87]. This led to the hypothesis that mutations in MYOF could lead to inflammation and edema through VEGF misregulation [87].

##### OTOF Mutations: Deafness, Autosomal Recessive 9 (DFBN9) and Auditory Neuropathy, Autosomal Recessive 1 (ANSD)

Mutations in *OTOF* cause a form of nonsyndromic autosomal recessive sensorineural deafness (DFBN9) resulting in congenital or prelingual bilateral deafness [88]. This phenotype is one of several that fall under the umbrella of auditory neuropathy spectrum disorder (ANSD), which includes a range of hearing dysfunctions [89]. *OTOF* mutations were the first identified genetic cause of ANSD, but they are not the only cause [89,90]. *OTOF* is expressed in cochlear inner hair cells and acts as a calcium sensor [91]. It functions at the presynaptic site to tether glutamatergic synaptic vesicles to the plasma membrane and plays a role in triggering their fusion and the replenishment of vesicles at ribbon synapses [89,92,93]. Mutations in *OTOF* disrupt neurotransmitter release at the ribbon synapses of sensory cells resulting in impaired signal transmission to the auditory nerve [94]. As a result, patients have severe-to-profound hearing loss likely from altered transmission of the auditory signal from the synapse to the brain [91]. Some pathogenic variants in *OTOF* can also cause a less frequent temperature-sensitive version of ANSD (TS-ANSD) [91]. Patients with TS-ANSD have normal-to-moderate hearing loss at baseline body temperature that turns to severe-to-profound hearing loss when the body temperature increases by at least 0.5 °C [95,96].

DFBN9 is associated with p.Tyr730Ter, p.Glu747Ter, p.Gln829Ter, p.Arg1792His, and p.Pro1825Ala mutations in OTOF. The p.Tyr730Ter mutation results in severe-to-profound hearing loss from a predicted truncated protein of 729 amino acids [97]. The p.Glu747Ter nonsense mutation also results in a truncated protein with the same diagnosis [98,99]. The p.Gln829Ter mutation is the most frequent OTOF mutant [100]. Patients with this mutation respond well to cochlear implantation, resulting in satisfactory functioning of the auditory nerve [101]. The p.Arg1792His is categorized as likely pathogenic using a manual classification system that includes data from non-public databases [102,103]. The p.Pro1825Ala was the first missense mutation identified in the *OTOF* gene and mutates a conserved proline residue in a C2 domain that is expected to bind calcium [104].

Auditory neuropathy is associated with the p.Pro50Arg, p.Ile515Thr, p.Gly541Ser, p.Glu551fs, p.Leu1011Pro, and p.Arg1939Gln mutations in OTOF. The p.Pro50Arg mutation is hypothesized to be “pathologic” because the amino acid is conserved in mouse OTOF; however, there are no experimental data to support this contention [105]. The p.Ile515Thr mutation is associated with a temperature-sensitive hearing phenotype and occurs in an amino acid conserved in human, mouse, chicken, and zebra fish [106]. The mutation occurs in the third C2C domain that is predicted to bind calcium [100]. This mutation is predicted to disrupt the structure of the third C2C domain and create a new myristylation site [107]. Mice homozygous for the mutation have a moderate hearing impairment and OTOF levels are significantly reduced, which results in enlarged synaptic vesicles and a reduction in exocytosis during stimulation [92]. The p.Gly541Ser mutation is hypothesized to be damaging because of the change of a non-polar residue in the C2C domain [108]. The p.Glu551fs mutation is predicted to result in protein truncation or nonsense-mediated decay [105]. The p.Leu1011Pro mutation is hypothesized to be pathogenic because the amino acid mutated is conserved and is in the fourth C2D domain that is thought to bind calcium [109]. The p.Arg1939Gln mutation is associated with a strong phenotype, occurs in a conserved amino acid (including FER-1, DYSF, and MYOF), and is predicted to be “damaging” to the protein from SIFT and Polyphen-2 analysis [103,110].

### 2.2. C. elegans MIB-1/FER-2/FER-4/SPE-16 and H. sapiens MIB1

#### 2.2.1. Ankyrin Repeat Domain (ANK) and RING Domain Function

*C. elegans mib-1/fer-2/fer-2/spe-16* and human *MIB1* encode MIB (mind bomb) E3 ubiquitin protein ligase 1 proteins [111,112,113,114]. *C. elegans mib-1* is orthologous to both human *MIB1* and *MIB2* [112]. Ubiquitin is conjugated to target proteins by a pathway including three enzymes: ubiquitin activation (E1 enzymes), ubiquitin conjugation (E2 enzymes), and ubiquitin ligation to specific substrates (E3 enzymes) [115]. MIB E3 ubiquitin protein ligase proteins ubiquitinate Notch ligands, including Delta [111,116]. Both human MIB-1 and *C. elegans* MIB-1 contain multiple ankyrin repeat domains (ANKs) and RING domains (Figure 2). Human MIB1 has nine ANK domains, while *C. elegans* MIB-1 contains seven [51,114]. ANK domains are approximately 33 amino acids long and are present in a plethora of proteins with a wide variety of functions [117,118]. ANK domains largely mediate protein–protein interactions, and many ANK domain containing proteins are necessary for ubiquitination [118,119]. ANK domains do not require highly conserved residues at specific sites to function, and instead the 3D structural fold is most likely responsible for their function, with variations in the exposed residues enabling specific protein binding [119]. Both orthologs have two RING domains, but human MIB1 also contains a third [51,114]. The RING (really interesting new gene)-type E3 ligases make up one of the three classes of ubiquitin transfer E3s [120]. The RING domain is defined by the cysteines and histidines that coordinate two zinc ions [121]. The RING domains in the MIB1 orthologs are the C3HC4 (Cys(3)-His-Cys(4)) RING/RING-HC type [51]. Recent studies have demonstrated that the third RING domain of MIB1 is the only RING domain that is necessary for autoubiquitination [122].

In addition to the ANK and RING domains, human MIB1 contains two MIB/HERC2 domains, a ZZ zinc finger domain (Znf), and two mind bomb SH3 repeat (REP) domains in the N-terminal half of the protein [51]. The two MIB/HERC2 domains and the ZZ zinc finger domain (Znf) are collectively referred to as the MZM element [123]. The MZM element and REP domains are necessary for interacting with Notch ligands, and they have independent substrate recognition regions [111,123,124].

#### 2.2.2. MIB-1 Function in *C. elegans*

The original characterization of phenotypes in *mib-1* arose from analyzing *fer-2* and *fer-4* mutants that were fertilization defective and originally identified in an EMS mutagenesis screen [54,55,113]. Sperm from these mutant males had “large tubules around their condensed chromatin” in the perinuclear region instead of the “amorphous RNA containing material” normally in this area [54]. *mib-1* mutants were also previously described as *spe-16* mutants, and additional alleles were identified in EMS and ENU mutagenesis screens [114].

*mib-1* hermaphrodites have severely reduced fertility at 25 °C, and sperm show a temperature-sensitive spermatogenesis defect [113,114]. MIB-1 is highly expressed in the germline, with the highest expression in the meiotic region [113,114]. No MIB-1 is seen in the spermatids, and it is instead packaged in the residual body (RB) [113,114]. In *C. elegans*, spermatids bud from an anucleate residual body after the completion of meiosis, and these spermatids will mature into fertilization-competent sperm after sperm activation (Figure 1) [22].

Most phenotypic characterization has been completed in *mib-1(eb154)* deletion mutants. Hermaphrodite *mib-1(eb154)* mutants show some defects in spermatogenesis and produce large spermatocyte-like cells with multiple haploid nuclei in addition to some normal spermatids [114]. Sperm from *mib-1(eb154)* have a variety of spermatogenesis defects including meiosis defects that result in abnormal spermatids and sperm and pseudopod defects [114]. *mib-1* mutants are also able to suppress gain-of-function mutants in two members of the LIN-12/Notch family of proteins, *lin-12* and *glp-1* [114]. Seven of the eight mutations highlighted are missense mutations (p.Met1Ile, p. Gly21Asp, p.Ala224Thr, p.Leu247Phe, p.Met291I, p.Leu292Gln, p.Cys720Tyr) and one is a nonsense mutation (p.Trp460Ter) (Figure 3).

#### 2.2.3. *mib-1* Orthologs Associated with Human Disease

##### MIB1 Mutations: Left Ventricular Noncompaction (LVNC) 7

Left ventricular noncompaction (LVNC) is a cardiomyopathy that affects the left ventricle of the heart, characterized by increased myocardial trabeculations [125,126]. Trabeculae are sheet-like protrusions of myocardium into lumen [127]. Instead of a compacted myocardium, the myocardium in LVNC has two layers: a thick “sponge-like layer” that contains intertrabecular spaces that communicate with the ventricles, and a thin subepicardial layer [125,126]. The development of LVNC is not well-understood [125]. There are 11 genetic causes of LVNC identified in OMIM; LVNC7 is caused by mutations in MIB1 [30].

The two mutations in MIB1 that are highlighted are p.Arg530Ter and p.Val943Phe (Figure 3). In patients with the p.Arg530Ter mutation, the truncated form of MIB1 was not detected in blood, indicating that the mutant mRNA may be degraded by nonsense-mediated decay. Both mutations affect MIB1 homodimer formation and ubiquitin ligase activity [128]. Neither mutant is able to ubiquitinate JAG1, which leads to defects in Notch signaling [128]. When both mutations were introduced into zebrafish, they led to defective cardiovascular development [128]. Both mutations result in a loss of function. In the p.Arg530Ter mutation, the loss of function is due to haploinsufficiency, and in the p.Val943Phe mutation, it is proposed to function as a dominant-negative version of the protein [128]. MIB1 mutations are thought to lead to LVNC through its interactions with NOTCH1 [128]. Without myocardial MIB1 activity, NOTCH1 in the endocardium is not activated by JAG1, and without NOTCH1 activation, the downstream events that are necessary for trabecular patterning and compaction cannot occur at the proper time during development [128].

### 2.3. C. elegans SPE-5 and H. sapiens ATP6V1B1 and ATP6V1B2

#### 2.3.1. Vacuolar ATPase Protein Function

SPE-5, ATPase H+ transporting V1 subunit B1 (ATP6V1B1), and ATPase H+ transporting V1 subunit B2 (ATP6V1B2) are Vacuolar-ATPase (V-ATPase) proteins (Figure 4). V-ATPases are large multi-subunit complexes originally characterized in the yeast vacuole [129,130]. The primary function of the V-ATPase is to create an electrochemical proton gradient across eukaryotic cell membranes, which is necessary for ATP generation [130]. The gradient created allows for the acidification of intracellular vesicles and organelles, which is necessary for many essential cell biological events to occur [130]. ATPase is composed of a cytosolic V_1_ domain and a transmembrane V_0_ domain [129]. The V_1_ domain is composed of eight different subunits (A, B, C, D, E, F, G, H) [129]. Mammalian gene names begin with the designation ‘*ATP6*’ and then have ‘V1’ or ‘V0’ for the domain followed by the subunit and isoform [129]. The B1 subunit is expressed in renal, epididymal cells in humans, while B2 expression is ubiquitous [129,131].

#### 2.3.2. SPE-5 Function in *C. elegans*

*spe-5* encodes one of two V-ATPase B-subunit paralogs in *C. elegans* [132]. *spe-5(ok1054)* deletion mutants cannot undergo MO acidification and form arrested, multi-nucleate spermatocytes [132]. The acidification of MOs is associated with MO morphogenesis and occurs at the same time as spermatid budding from the residual body and FB-MO (fibrous body–membranous organelle—see Figure 1) maturation into an MO secretory vesicle [132]. SPE-5 expression begins in the spermatocytes [132]. The majority of SPE-5 is discarded into the residual body; however, some SPE-5 is maintained in MOs of budded spermatids [132]. *spe-5* is on Chromosome I and is upregulated during spermatogenesis. In contrast, the paralog of *spe-5*, *vha-12*, is present on the X chromosome, which is usually silenced during spermatogenesis [132]. Expression of *vha-12* via extrachromosomal array in *spe-5* mutants can lead to the production of small broods, indicating the functional equivalence of the two genes [132]. Three of the highlighted mutations are missense mutations (p.Asp172Asn, p. Gly204Arg, and p. Pro382Ser) (Figure 4). The p.Asp172Asn and p.Pro382Ser mutations occurred in conserved amino acids [132]. The fourth mutation (p.Trp473Ter) is a nonsense mutation [132].

#### 2.3.3. spe-5 Orthologs Associated with Human Disease

##### ATP6V1B1 Mutations: Distal Renal Tubular Acidosis with Progressive Sensorineural Hearing Loss (DRTA2)

Mutations in the *ATP6V1B1* gene, encoding the B subunit of the apical proton pump mediating distal nephron acid secretion, cause one of the four types of distal renal tubular acidosis with progressive bilateral sensorineural hearing loss (DRTA2) [30,133,134]. Clinical manifestations of abnormal renal function occur in pediatric patients during childhood and include failure to thrive, dehydration, vomiting, polyuria, hypocitraturia, and nephrocalcinosis [135]. ATP6V1B1 is expressed in the cochlea and endolymphatic sac, where it may be necessary to maintain the pH of the endolymph [133,136].

The nonsense mutation in arginine 31 (p.Arg31Ter) and the frameshift mutation at threonine 166 (p.T166fs), which results in premature termination at 174, both likely result in a loss of function [133]. The mutation of leucine 81 to proline occurs in a conserved residue and is predicted to disrupt an N-terminal ß-barrel [133]. The leucine 81 residue is conserved in both ATPGV1B2 and *C. elegans* SPE-5 using ClustalW multiple sequence alignment. In co-expression studies, the missense mutations p.Gly78Arg and p.Leu81Pro (as well as other missense mutations not highlighted here) fail to interact with the V-ATPase E subunit, indicating that the mutations could impair ATPase complex formation [137].

##### ATP6V1B2 Mutations: Dominant Deafness–Onychodystrophy (DDOD) Syndrome and Zimmermann–Laband Syndrome (ZLS)

A heterozygous nonsense mutation in *ATP6V1B2* has been reported in dominant deafness-onychodystrophy syndrome (DDOD) [138]. DDOD syndrome is characterized by congenital sensorineural deafness and onychodystrophy [131]. The presentation of onychodystrophy is variable, with absent or hypoplastic fingernails and/or toenails, bulbous fingertips, or pyramidal-shaped distal phalanx of the toes [139]. The p.Arg506Ter mutation is a haploinsufficient mutation that occurs in a highly conserved amino acid and results in a truncated protein, which is predicted to prevent a hydrogen bond from forming between tyrosine 504 and aspartic acid 507 [140]. In transfected cells, the p.Arg506Ter mutation results in reduced lysosomal acidification [140].

Zimmermann–Laband syndrome (ZLS) is a rare disorder characterized by gingival fibromatosis, nail aplasia or hypoplasia, joint hyperextensibility, hepatosplenomegaly, hirsutism, abnormalities of the cartilage of the nose and/or ears, and intellectual disability [138]. There are three types of ZLS, and Zimmermann–Laband syndrome-2 (ZLS2) is caused by mutation in the *ATP6V1B2* gene [30]. Epilepsy is also common in ZLS1 but variable in ZLS2 [141]. Arg485 is highly conserved, including in *C. elegans*, and the p.Arg485Pro mutation is not predicted to have any impact on the affinity for its substrate [138].

There is partial clinical overlap between ZLS and DDOD, and ZLS- and DDOD-causing *ATP6V1B2* mutations affect the same region of the protein, which is involved in the formation of the V1 subcomplex assembly [138].

### 2.4. C. elegans SPE-9 and H. sapiens DLL1 and DLL4

#### 2.4.1. EGF-like Domain Function

The proteins encoded by *spe-9* and its orthologs *Delta-like canonical Notch ligand 1* (*DLL1*) and *Delta-like canonical Notch ligand 4* (*DLL4*) contain multiple EGF-like domains (Figure 5). According to InterPro, SPE-9 contains nine EGF-like domains, with four capable of binding calcium [51]. In previous models, *spe-9* was shown to encode ten EGF-like repeats [142]. DLL1 and DLL4 both contain eight EGF-like domains, with six EGF calcium-binding domains [51]. Proteins containing EGF-like domains are commonly involved in extracellular events like adhesion and receptor–ligand interactions [143]. EGF calcium-binding repeats are often found in conjunction with non-calcium-binding repeats [144]. Calcium may be used to coordinate the interactions between proteins [145]. EGF-like domains may also be used as a scaffold for functions or a spacer unit on cell-surface proteins [146]. The role of EGF-like domains can vary. For example, only two of the 36 EGF-like domains of Notch are necessary and sufficient for Notch–Delta interaction, and in *C. elegans*, the EGF-like domains in LAG-2 are not necessary for its function when binding to the LIN-12/Notch family members LIN-12 and GLP-1 [143,147].

In addition to multiple EGF domains, DLL1 and DLL4 both contain DSL (Delta, Serrate, lag-2) and MNNL (module at the N-terminus of Notch ligands)/C2 domains. The DSL domain is required for Notch trans-activation or cis-inhibition [148]. The MNNL domain is also referred to as a C2 domain and is a membrane recognition motif involved in binding phospholipids [149,150,151]. Mutations in EGF repeats near the MNNL-DSL binding interface have been shown to change the DLL1 function, indicating that the protein context of the domains is important for protein function [152].

#### 2.4.2. SPE-9 Function in *C. elegans*

In *C. elegans, spe-9* was the first gene to be identified as necessary for fertilization [142]. Sperm from *spe-9* mutants can contact oocytes but cannot fuse with them, despite having normal sperm morphology and motility [26,142]. SPE-9 is localized in the pseudopod of mature *C. elegans* sperm [153]. Mutations in even one of EGF-like domains result in infertility, and the EGF domains must be tethered to the sperm surface to be functional [154]. There are two temperature-sensitive alleles that result in missense mutation, with *spe-9(hc52)* containing a p.Gly550Glu mutation and the *spe-9(hc88)* allele having a p.Cys258Tyr mutation [142]. Cysteine 258 in the third EGF-like repeat is conserved [142]. Singson et al. reported that glycine 550 in the eighth EGF-like repeat is conserved [142]. However, a ClustalW multiple sequence alignment with DLL1 and DLL4 shows no conservation at that residue. Two nonconditional alleles, *spe-9(eb19)* and *spe-9(eb23)*, have nonsense mutations in glutamic acid 151 (p.Glu151Ter) and arginine 504 (p.Arg504Ter) [142].

#### 2.4.3. *spe-9* Orthologs Associated with Human Disease

##### DLL1 Mutations: Neurodevelopmental Disorder with Nonspecific Brain Abnormalities and with or Without Seizure (NEDBAS)

The features of neurodevelopmental disorder with nonspecific brain abnormalities and with or without seizure (NEDBAS) include intellectual disability, autism spectrum disorder, seizures, variable brain malformations, muscular hypotonia, and scoliosis [155,156]. DLL1 is one of the ligands for the Notch receptor [156]. The Notch signaling pathway coordinates developmental processes in most organs and tissues [157]. DLL1 has been shown to be important in neurodevelopment [155]. Three of the highlighted mutations in *DLL1* result in nonsense mutations (p.Cys77Ter, p. Glu498Ter, and Arg 509Ter) and the fourth is a frameshift mutation in glutamic acid 673 (p.Glu673Glyfs*15) [155]. The precise impact of these mutations on DLL1 function is unknown.

##### DLL4 Mutations: Adams–Oliver Syndrome 6 (AOS6)

Patients with Adams–Oliver syndrome (AOS) present with aplasia cutis congenita (congenital scalp defects) and terminal transverse limb defects that vary in severity [158]. There are six forms of Adams–Oliver syndrome defined by the gene that is mutated, and AOS6 is defined by a mutation in *DLL4* [30]. *DLL4*-associated AOS shows highly variable clinical features [159]. DLL4 is a ligand for Notch receptors and is essential for vascular formation [160,161]. The p.Ala121Pro is predicted to disrupt the structure and function of the beta-strand on the inside of the module at the N-terminus of the Notch ligand (MNNL) domain [159]. The p.Arg186Cys mutation could alter the conformation of the protein by introducing a novel disulfide bond and therefore have a negative impact on the DSL domain function [159]. The cysteine-replacing mutations at amino acid 390 (p.C390R and p.C390Y) affect residues in the fifth EGF-like domain, which are conserved up to *D. melanogaster* [159]. A ClustalW multiple sequence alignment shows that cysteine 390 is conserved in DLL1 but not *C. elegans* SPE-9. The final two highlighted mutations (p.Gln554Ter and p. Arg558Ter) are nonsense mutations that are predicted to result in nonsense-mediated decay of the mRNA transcript [159].

### 2.5. C. elegans SPE-15 and H. sapiens MYO6

#### 2.5.1. Myosin VI Protein Function

SPE-15 and myosin VI (MYO6) are myosin domain containing proteins (Figure 6). Myosins are molecular motors responsible for actin-based motility [162]. Myosin heavy chains have three main regions: the motor head, the neck region (lever), and the tail domain (cargo-binding domain) [162]. The motor head domain contains the nucleotide binding pocket and the actin binding site [163]. The neck region contains a variable number of isoleucine–glutamine (IQ) motifs and acts as a lever arm to modulate step size [163]. The tail region can vary greatly and is also referred to as the cargo-binding domain (CBD) [163]. The MYO6 tail region contains short coiled-coil regions to mediate myosin dimerization, helical regions to extend the length of the lever arm, and “single alpha-helix (SAH) regions that also serve as lever arm extensions” [163]. There are both conventional and unconventional myosins [163]. Class II myosins, including skeletal muscle myosin-2, are referred to as ‘conventional’ myosin [164]. All other myosins are considered ‘unconventional’ [18,163,164,165]. The unconventional myosins perform a wide range of cellular processes, including trafficking, membrane tension, and cell motility [165]. SPE-15 and MYO6 are unconventional myosins, which are the only myosin types known to move toward the pointed (-) end of actin filaments [18,165,166,167]. *MYO6* is regulated by alternative splicing depending on the cell and tissue type [163]. According to the SMART database, both SPE-15 and MYO6 contain one IQ domain in the neck region and a coiled-coil and cargo-binding domain in the tail region [168].

#### 2.5.2. SPE-15 Function in *C. elegans*

*spe-15* is necessary for the separation of spermatids from RBs [169]. SPE-15 works with actin to create an actomyosin ring that constricts the membrane between the spermatid and RB [169]. During spermatid budding from the RB, SPE-15 migrates to the site of membrane constriction from the spermatid poles, indicating that it plays a role in membrane constriction and spermatid release [169]. SPE-15 is necessary for partitioning organelles, including mitochondria, ribosomes, and cytoskeletal components, into the spermatids and away from the residual body, and proper organelle segregation is required to deliver the necessary components for spermatid activation and sperm function [18,169,170]. As a result of improper organelle segregation, *spe-15* mutant spermatids individualize but have defects in sperm activation [18,169]. *spe-15* alleles have varied responses to sperm activators in vitro. *spe-15(ok153)* and *spe-15(qx529)* have defects in sperm activation, while *spe-15(hc75)* does not have activation defects [18,26,169]. *spe-15(ok153)* is a deletion mutation that removes sequences that encode the myosin head motor domain [18]. *spe-15(qx529)* is a CRISPR/Cas9-generated allele designed to mutate the amino acids in the “RRL” motif (amino acids 1067-1069) within the cargo-binding domain (CBD) to a triple-alanine mutation (p.Arg1067Ala, p.Arg1068Ala, and p.Leu1069Ala) [169]. RRL motifs in the CBD can bind multiple adaptor proteins to regulate and coordinate motor–cargo attachment [171]. According to the InterPro annotation, the RRL motif amino acids are outside of the CBD domain [51]. Regardless, in *spe-15(qx529)* mutants, the cargo adapter GIPC (RGS-GAIP-interacting protein C terminus) interactions are disrupted, and spermatid release cannot occur normally [169]. *spe-15(hc75)* is a nonsense mutation in tryptophan 804 (p.W804Ter) [18,26]. The inability of sperm to activate is the most likely cause of infertility in *spe-15* mutants [18,169].

#### 2.5.3. *spe-15* Orthologs Associated with Human Disease

##### MYO6 Mutations: Deafness (Autosomal Dominant and Recessive), Including Hypertrophic Cardiomyopathy

Mutations in *MYO6* are associated with deafness. The p.Cys442Tyr and p.Arg849Ter mutations are associated with autosomal dominant nonsyndromic sensorineural hearing loss [172]. The hearing loss onset in these cases varied from young, early-school-age patients to adults and often could evolve to profound deafness [173,174,175]. The p.Cys442Tyr mutation occurs in the motor domain in a highly conserved position [173]. Mechanoenzymatic analysis showed that MYO6 proteins containing this mutation have accelerated ADP dissociation rates and decreased processive movement [176]. *C. elegans* have a similar hydrophilic amino acid (serine) at this position [173]. The p.Arg849Ter mutation occurs in the lever arm domain of MYO6 (in between the IQ and coiled-coil domains in alternative models) and results in a truncated protein missing the globular cargo-binding domain [174,177,178]. This truncated version of MYO6 is hypothesized to disrupt dimerization, cargo binding, and localization [174]. In some cases, deafness co-segregated with familiar hypertrophic cardiomyopathy, as in p.His246Arg mutations [179]. Histidine 246 is also conserved in *C. elegans*, as determined by a ClustalW multiple sequence alignment. The p.Arg1166Ter mutation occurs in the globular domain of the tail region [172]. Finally, *MYO6* autosomal recessive mutations (p.Thr13fs, p.Glu216Val, and p. Arg1166Ter) have also been reported to result in bilateral, profound, congenital sensorineural hearing loss, which included vestibular dysfunction and mild facial dysmorphism [172]. The p.Thr13fs mutation (36-37insT) is predicted to cause a premature translation termination after the first 12 amino acids of MYO6 [172]. The glutamate mutated in the p.Glu216Val mutation is conserved in MYO6 proteins in mouse and other animals, and in *C. elegans*, there is an aspartate at this residue, indicating that charge may be important for function [172]. The p.His246Arg mutation occurs in a conserved histidine of the motor domain region in multiple species, including mouse and *Drosophila* [179].

### 2.6. C. elegans SPE-26 and H. sapiens KLHL10

#### 2.6.1. Kelch Repeat Domain Function

Both SPE-26 and Kelch-like family member 10 (KLHL10) contain multiple Kelch repeat domains (Figure 7). Proteins containing Kelch repeat domains participate in a range of cellular functions including extracellular communication, cell morphology maintenance, and regulation of gene expression [180]. Kelch domains typically interact with Cullin3 to form E3 ligase complexes, which mediate the ubiquitination of substrate proteins [180]. Each Kelch repeat folds to form a single blade of a β-propeller structure, and the number of blades varies depending on the number of repeats [181]. Several Kelch repeat proteins associate with the actin cytoskeleton, and the β-propeller is important for function [181]. Many Kelch-like homologs (KLHL) contain a BTB/POZ (BTB for Broad-complex, tramtrack, and bric à brac and POZ for Pox virus and Zinc finger) domain, a BACK (BTB and C-terminal Kelch) domain, and multiple Kelch repeat motifs [180,182]. The BTB domain recruits E3 ubiquitin ligase complexes, and BACK domains form a linker between the BTB/POZ domain and the Kelch repeats [180,182]. The presence of a BTB/POZ domain and BACK domain and the number of Kelch repeats can vary in SPE-26 protein models [51]. Figure 7 is the model from the SMART database and indicates that KLHL10 contains a BTB/POZ domain, a BACK domain, and six Kelch repeats [168]. SPE-26 is a member of the C-propellor (C-prop) structural subgroup of Kelch repeat proteins, which contain five to seven Kelch motifs within the C-terminal region but have no other regions of homology within their primary sequence [181]. KLHL10 is a member of the N-dimer, C-propeller subgroup because of the BTB/POZ domain in the N-terminal region [181].

#### 2.6.2. SPE-26 Function in *C. elegans*

Hermaphrodite and male *spe-26* mutants are sterile because of spermatogenesis disruptions [183]. Primary spermatocytes develop normally but fail to complete meiosis and do not form haploid spermatids [183]. The FB-MO appears normal in primary spermatocytes in *spe-26* mutants but disassembles prematurely in the spermatocyte instead of the spermatid, as normally occurs, and the arrays of major sperm protein (MSP) disperse [183]. The MO membranes are able to fold into the compact dumbbell shape of a mature MO, but in some spermatocytes, the MO membranes are distended [183]. Sperm from *spe-26* mutants also have multiple nuclei and show mislocalized actin filaments, ER, and ribosomes [183]. Normally, actin, ER and ribosomes are sequestered into the residual body; however, this does not occur in *spe-26* mutants [183]. There are six *spe-26* alleles, and five of the six *spe-26* alleles map within Kelch repeats [181]. *spe-26(eb8)* and *spe-26(hc139)* are nonconditional alleles resulting from a nonsense mutation (glutamine 441 and tryptophan 460, respectively) in the fifth Kelch repeat [183]. *spe-26(hc140)* has a mutation (p.Gly446Glu) in a conserved glycine in the fifth repeat unit of blade five [183]. *spe-26(it112)* and *spe-26(it118)* are two independently isolated alleles with the same base substitution, resulting in a missense mutation (p.Ser360Asn) in the third repeat element [183]. The sixth allele, *spe-26(hc138)*, is a temperature-sensitive mutation in the splice site in the second intron and is not shown in Figure 5 [183].

#### 2.6.3. *spe-26* Orthologs Associated with Human Disease

##### *KLHL10* Mutations: Spermatogenic Failure 11

Mutations in *KLHL10* result in oligozoospermia, characterized as spermatogenic failure [184]. Spermatogenic failure displays wide genetic heterogeneity, with 87 different mutations cataloged in OMIM [30]. In cases of spermatogenesis failure, spermatogenetic arrest during meiosis causes infertility [185]. The cause of infertility in patients with meiotic arrest is often unidentified; however, in general, the patients display meiotically-arrested spermatocytes in the seminiferous tubules [186]. The two missense mutations highlighted in KLHL10 (p.Gln216Pro and p.Ala313Thr) prevent homodimerization in yeast two-hybrid assays and are hypothesized to affect the function [184]. In mice, *Klhl10* mRNA is expressed in steps 1-16 spermatids, and the protein is detected in the cytoplasm of steps 9-16 spermatids [187]. *Klhl10* haploinsufficient males have defects in late spermiogenesis, possibly from defects in ubiquitination-regulated protein turnover [187].

### 2.7. C. elegans SPE-39 and H. sapiens VIPAS39

#### 2.7.1. VIPAS39/SPE-39 Protein Function

The human ortholog of *spe-39* is *VPS33B interacting protein, apical-basolateral polarity regulator, spe-39 homolog (VIPAS39*), also known as C14orf133, Vps16b, and VIPAR [188,189]. *spe-39* gene function was originally discovered in *C. elegans*, and subsequent studies showed there was an interaction between VPS33B and SPE-39 [190,191]. Despite being orthologs, SPE-39 and VIPAS39 have different domain structures (Figure 8). VIPAS39 contains a VPS16 C-terminal domain [29,51]. VPS16 forms part of a complex that is required for vacuolar biogenesis and vacuolar stability [192]. The C-terminal portion of VPS16 binds to VPS33 and is necessary for VPS33 to be integrated into a multi-subunit complex called HOPS (Homotypic fusion and vacuolar protein sorting) [193]. The HOPS complex is necessary for tethering and fusion of lysosomes and intracellular compartments [194]. Human SPE-39 interacts with VPS33 homologs and other HOPS subunits [191]. *C. elegans spe-39* is not predicted to encode any conserved protein domain according to the SMART and InterPro databases [51,168]. Zhu and L’Hernault identified a unique motif in *C. elegans spe-39* and additional orthologs [(LM)-(ED)-x-(FY)-(RK)-S-x-x-(DE)-K-x-x-L-L-x-x-(AL)-(VIM)] [190]. However, that motif is not cataloged in InterPro or the conserved domains database (CDD) [51,195].

#### 2.7.2. SPE-39 Function in *C. elegans*

*spe-39* mutants complete meiosis but do not form spermatids, possibly due to issues with vesicular biogenesis [190]. There are four *spe-39* mutants: *spe-39(eb9)*, *spe-39(eb110)*, *spe-39(eb111)*, and *spe-39(tx12)*. *spe-39(eb9)*, *spe-39(eb110)*, and *spe-39(eb111)* are nonsense mutations, at amino acid positions 334 (p.Arg334Ter), 320 (p.Gln320Ter), and 393 (p.Arg393Ter), respectively [190]. *spe-39(tx12)* is a splice-site mutant and is not depicted in Figure 8 [190]. Hermaphrodites bearing any of these mutant alleles are self-sterile and can produce progeny after crossing with either wild-type or heterozygous *spe-39* mutant males [190]. *spe-39*(eb9) and *spe-39(tx12)* mutants have dramatic reductions in spermatogenesis; their sperm do not contain MOs and have disorganized FBs that are not surrounded by membrane [190]. In vitro, SPE-39 interacts with two homologs of the vacuolar protein-sorting protein VPS33 (VPS33A and VPS33B) [191]. RNAi interference of *VPS33B* in *C. elegans* causes *spe-39*-like defects, including abnormal FB-MO morphogenesis and cytokinesis [191].

#### 2.7.3. *spe-39* Orthologs Associated with Human Disease

##### VIPAS39 Mutations: Arthrogryposis, Renal Dysfunction, and Cholestasis 2 (ARC2)

Arthrogryposis (multiple joint fractures or stiffness), renal dysfunction, and cholestasis (decrease in bile flow) (ARC) syndrome is a rare multisystem disorder with wide clinical variability [196]. Patients diagnosed with ARC syndrome also have low g-glutamyl transpeptidase activity [197,198]. Fanconi syndrome, ichthyosis, dysmorphism, jaundice, and diarrhea are also associated with ARC syndrome and are part of the spectrum of phenotypes [199]. ARC2 is caused by mutations in *VIPAS39*, while ARC1 is caused by mutations in *VPS33B* [30]. Patients with mutations in *VIPAS39* display a typical ARC phenotype [200,201]. The majority of the mutations outlined here (p.Trp59Ter, p.Gln179Ter, p.Thr250fs, p.Gln291Ter, and p.Arg381Ter) are predicted to result in an incomplete, truncated protein being produced because of the insertion of a nonsense codon [200,201]. The p.Met1Arg mutation is predicted to result in translation failure [201]. Tornieri et al. showed that the p.Arg220Ter nonsense mutation led to a loss of VIPAS39 binding to VPS33B in yeast two-hybrid assays [202].

### 2.8. C. elegans UBA-1/SPE-32 and H. sapiens UBA1

#### 2.8.1. Ubiquitin-Activating Enzyme Protein Function

Both *C. elegans uba-1* and human *ubiquitin- like modifier activating enzyme 1* (*UBA1*) are ubiquitin- activating enzymes (Figure 9). UBA1 is a ubiquitin-activating enzyme (E1) [203,204]. The UBA1 structure has a relatively complex arrangement of multiple domains [205]. The adenylation domains composed of two MoeB/ThiF-homology motifs, which may be illustrated in alternative representations, are not shown in Figure 9 [205]. Instead, in Figure 9, we have highlighted the catalytic cysteine half-domains embedded within the MoeB/ThiF motifs [51]. The catalytic cysteine half-domains contain the E1 active site cysteine, referred to as the first catalytic cysteine half-domain (FCCH), and the second catalytic cysteine half-domain (SCCH) [205]. The FCCH domain plays a role in ubiquitin recognition, while the SCCH domain contains the catalytic residue involved in thioester bond formation [205,206]. UBA1 also contains a four-helix bundle (4HB) that is immediately adjacent to the C-terminal of the FCCH domain [205]. The FCCH and 4HB domains are part of the “inactive” adenylation domain, while SCCH is part of the “active” adenylation domain that binds ATP and ubiquitin [205]. Finally, UBA1 contains a C-terminal ubiquitin-fold domain (UFD), which recruits and binds E2 ubiquitin-conjugating enzymes [205,207].

#### 2.8.2. UBA-1 Function in *C. elegans*

UBA-1 is the only ubiquitin-activating enzyme in *C. elegans* [208]. RNAi depletion of *uba-1* results in embryonic lethality, a low brood size, and death [209]. A temperature-sensitive allele *uba-1(it129)* was isolated based on sterility and larval lethality and provisionally designated as *spe-32*; however, *spe-32* was determined to be allelic to *uba-1* [210]. The *uba-1(it129)* strain contains a p.Pro1024Ser substitution, and the resulting phenotype is hypomorphic [210]. *uba-1(it129)* mutant adult hermaphrodites are incapable of fertilization and lay unfertilized oocytes [210]. Sperm from these hermaphrodites are incapable of fertilization, despite having a normal morphology and being produced in normal numbers [210]. Sperm from *uba-1(it129)* hermaphrodites have normal motility and localization; however, they are not maintained in the spermatheca as unfertilized oocytes pass through [210]. UBA-1 function in *C. elegans* sperm may be necessary for protein sorting during spermatid division from the residual body, similar to the role of *spe-15*, or ubiquitination may play a role in fertilization, as it does in ascidians [210,211].

#### 2.8.3. *uba-1* Orthologs Associated with Human Disease

##### UBA1 Mutations: VEXAS Syndrome, Somatic and Spinal Muscular Atrophy, X-Linked 2, Infantile (XL-SMA)

VEXAS (vacuoles, E1 enzyme, X-linked, autoinflammatory, somatic) syndrome describes a multitude of adult-onset inflammatory syndromes that were previously thought to be unrelated [212]. VEXAS syndrome is characterized by a spectrum of systemic inflammatory manifestations (relapsing polychondritis, Sweet’s syndrome, polyarteritis nodosa, or giant-cell arteritis) and hematologic symptoms (myelodysplastic syndrome or multiple myeloma) [212,213]. The syndrome effects multiple organ systems and is severe and progressive [213]. Recurrent fever and skin lesions are the most common clinical features; however, the spectrum of clinical characteristics has expanded as new cases have been reported since the original genotype-driven approach to defining the disease [212,213]. Somatic mutations in methionine 41 (p.Met41val, pMet41Thr, and p.Met41Leu) of *UBA1* were identified as causing VEXAS syndrome [212]. Patients with these mutations have vacuoles in myeloid and erythroid precursor cells, but the exact mechanism of vacuolation in VEXAS syndrome patients is unknown [212,213]. *UBA1* is expressed as two isoforms: UBA1a and UBA1b [212]. UBA1a initiates at p.Met1 and is localized to the nucleus, while UBA1b initiates at p.Met41 and is cytoplasmically expressed [212]. Mutations in p.Met41 result in the loss of cytoplasmic UBA1b, and instead lead to the expression of a third isoform initiated at p.Met67 (UBA1c) [212]. UBA1c was also expressed in the cytoplasm but was catalytically inactive in thioester assays [212]. An additional variant resulting in a serine to phenylalanine substitution (p.Ser56Phe) was also identified, which causes a temperature-sensitive reduction in UBA1 enzymatic activity [214]. One hypothesis for how UBA1 mutations cause VEXAS syndrome is that the mutations lead to the activation of the unfolded protein response (UPR) and multiple inflammatory pathways [215].

X-linked infantile spinal muscular atrophy (XL-SMA, SMAX2) is a rare neuromuscular disease defined by congenital hypotonia, arthrogryposis, bone fractures present at birth, neurogenic atrophy, denervation, anterior horn cell loss, and a shortened life span (less than two years) because of respiratory distress or pulmonary hypoplasia [216]. Large-scale mutation analysis identified two missense mutations (p.Met539Ile and p.Ser547Gly) and one synonymous substitution (c.1731 C/T, p.Asn577Asn) in *UBA1* that segregated with XL-SMA [217]. The mutations cluster in the active adenylation domain towards the N-terminus of the SCCH that is required for the catalytic activity of UBA-1 [218,219]. The missense mutations do not have any impact on the catalytic adenylation activity of UBA1 in vitro; however, another mutation not highlighted here (E557V) results in a reduction in adenylation activity [218]. The synonymous mutation p.Asn577Asn reduces *UBA1* expression and alters the methylation pattern of the exon it is within [217]. UBA1 may also play a role in more common *SMN1*-dependent spinal muscular atrophy [219,220].

## 3. Discussion

Employing *C. elegans* fertilization genes as a cell biology model and human disease model has led to significant contributions and advancements in the understanding of protein function and cellular function. This review has outlined the role of eight *C. elegans fer* and *spe* mutants and their human orthologs, and, of those, a few are of specific note. When mutations in *DYSF* were identified as causing LGMD2B, the only sequence they showed homology to was *C. elegans fer-1* [35]. The next year, when mutations in *OTOF* were identified as responsible for DFNB9, *OTOF* was only homologous to *fer-1* and the newly identified *DYSF* [97]. The studies of OTOF, and other ferlin family members, including *fer-1*, have contributed to the understanding of ferlin protein function and the development of gene therapies to cure hearing loss in individuals with *OTOF* mutations [221,222,223,224,225]. The initial characterizations of *C. elegans spe-39* were foundational for understanding how VIPAS39 functions, including that it interacts with VPS33B, and set the stage for understanding how human VIPAS39 functions and how vacuolar protein sorting is regulated [191]. In addition, mutations in both *C. elegans spe-26* and human *KLHL10* cause disruptions in sperm function, demonstrating a remarkable similarity in gene–phenotype relationships between species.

As we come to understand more about the gene–phenotype relationships in *C. elegans*, we may discover that the genes necessary for spermatogenesis and/or fertilization represent phenologs of human phenotypes. Phenologs are defined as the phenotype-level equivalent of gene orthologs and are evolutionarily conserved traits or defects that arise from a set of conserved genes [226]. The study of *C. elegans* has already identified phenologs of human disease, including amyotrophic lateral sclerosis (ALS), spinal muscular atrophy (SMA), and polycystic kidney disease (PKD) [227]. For example, an atypical form of ALS can be caused by mutations in *VAPB* (VAMP (synaptobrevin)-associated protein B), and mutations in one of the *C. elegans* orthologs VAPB, *vpr*-1, have defects in distal tip cell migration and abnormal positions of head neurons [228,229]. SMA is caused by decreased levels of the survival motor neuron (SMN) protein [230]. *C. elegans smn-1* mutants exhibit a reduced lifespan, larval arrest, and impaired locomotion and pharyngeal pumping [231]. Analysis of these mutants showed that SMN-1 is necessary for synaptic vesicle recycling [232]. Polycystic kidney disease is caused by mutations in either PKD1 or PKD2 [233]. *C. elegans pkd-2* is necessary for ciliated sensory neurons in the male tail to detect potential hermaphrodite mates, respond to contact, and locate the vulva [234,235,236]. The second *C. elegans* polycystin, *lov-1*, acts in the same pathway and has the same phenotype [234,235,236].

New opportunities for understanding will arise as new gene–phenotype relationships in both humans and *C. elegans* are identified and the functions of understudied and uncharacterized genes are elucidated. The *C. elegans* genes reviewed here were restricted to those with defined human orthologs that have phenotypes cataloged in OMIM. As more gene–phenotype relationships are established in humans, we will undoubtedly be able to expand the scope of a review such as this to potentially include the discussion of genes without a current phenotype relationship. One example of a potential new gene–phenotype relationship is the identification that *MIB2* variants are possibly associated with a gastropathy that resembles Ménétrier disease [237]. This relationship is not yet confirmed on OMIM, and the associated MIB2 variants are categorized as “Variant of unknown significance” (Table 1) [30]. *MIB2* is an ortholog of *C. elegans mib-1*, and studies of *mib-1* could give insights into MIB2 function [112,113]. Additionally, more *C. elegans* genes that are important for spermatogenesis and fertilization continue to be discovered, including two new sperm genes necessary for fertilization, *spe-36* and *spe-51* [238,239]. There is also another class of *C. elegans* genes on the egg that are necessary for fertilization—the *egg* genes—which could provide insights into human disease. *egg* mutants produce oocytes that cannot be fertilized and/or undergo egg activation to properly trigger the oocyte-to-embryo transition, and their infertility phenotype cannot be rescued by mating with wild-type males [240,241]. There are currently six categorized *egg* genes, with the search for more underway [242]. Of the six *egg* genes, *egg-4* and *egg-5* have orthologs in humans. The human orthologs of *egg-4* and *egg-5* are *PTPRG* and *PTPRZ1*; however, these genes have no associated gene–phenotype relationship currently [30]. Finally, there are also evolving methods of defining orthologs, including structure-based methods, which could alter our view of how we compare function across proteins.

Using *C. elegans* to determine the molecular function of pathogenic human variants, as well as variants of unknown significance, has the capability to aid significantly in precision medicine [33]. The molecular functions for most of the 64 human sequence variants discussed in this review have not been experimentally analyzed, and our knowledge of their impacts on protein functions is only based on the predictive impacts of the amino acid changes. Further studies of the molecular functions of these variants could impact treatment. For example, two *DYSF* variants can lead to substitutions in the same amino acid, with p.Gly299Trp being associated with MMD, while a p.Gly299Arg mutation is associated with LGMD2B [64]. The details of how the p.Gly299Arg mutation could lead to amyloid deposits in the muscle and whether p.Gly299Trp does the same are not understood. In addition, studying how variants impact protein function in a variety of protein domain architectures can provide insights into how protein domain organization impacts functions [243].

The genes and proteins reviewed here have led to significant advances in knowledge on human disease, protein domain function, and *C. elegans* reproduction, and they demonstrate the power of using model organisms to study human disease. Combining the study of human variants with additional studies that characterize the precise molecular functions of the existing *C. elegans* variants using the plethora of tools available will lead to a deeper understanding of cellular function. The processes of making gametes, getting them to the site of fertilization, cellular fusion, and triggering the development of a new organism are full of complex cellular processes that are ripe for study.

## Figures and Tables

**Figure 1 jdb-13-00004-f001:**
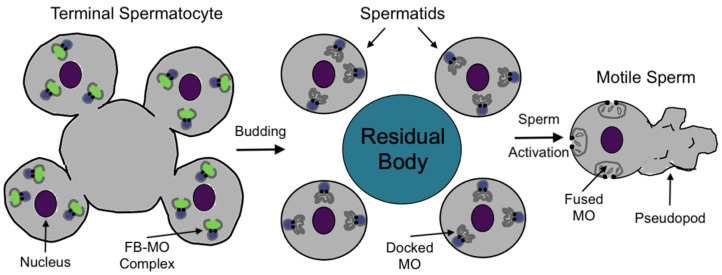
Key events of spermatogenesis and sperm activation. The terminal spermatocyte contains budding spermatids surrounding a developing residual body (RB) [16,21,22,23]. Spermatids contain lysosome-like fibrous body–membranous organelles (FB-MOs) [16,21,22,23]. In the FB-MO complex, the FB (fibrous body) is colored green, and the MO is colored blue. MOs are Golgi-derived and associate with FBs to form the FB-MO complex [16,21,22,23]. During spermatid differentiation, round spermatids bud from the terminal spermatocyte in an asymmetric division [16,21,22,23]. Spermatids selectively retain the FB-MOs and cellular components that are no longer needed are shed to the residual body [16,21,22,23]. The MO membrane also retracts and docks with the plasma membrane, and the FB is released and disassembled [16,21,22,23]. During sperm activation (spermiogenesis), the round spermatids extend a pseudopod and become motile, and the MO fuses with the plasma membrane and releases its contents [16,21,22,23]. The purple oval represents the nucleus in all stages.

**Figure 3 jdb-13-00004-f003:**
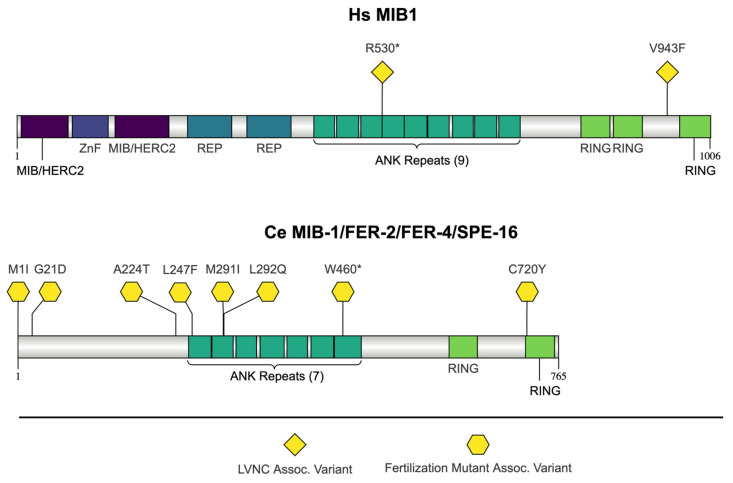
Protein diagram of *C. elegans* MIB-1/FER-2/FER-4/SPE-16 and *H. sapiens* MIB1. Diagram depicting full-length protein with protein domains and genetic variants. Human and *C. elegans* genetic variants are labeled with the shapes indicated at the bottom of the figure (asterisk (*) indicates a nonsense mutation). Schematic diagram was generated using Illustrator for Biological Sequences (IBS) 2.0 [49].

**Figure 4 jdb-13-00004-f004:**
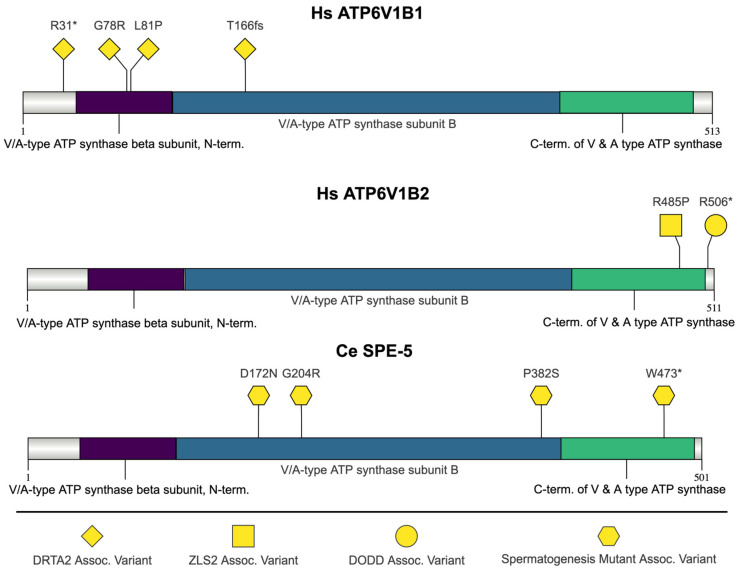
Protein diagram of *C. elegans* SPE-5 and *H. sapiens* ATP6V1B1 and ATP6V1B2. Diagram depicting full-length protein with protein domains and genetic variants. Human and *C. elegans* genetic variants are labeled with the shapes indicated at the bottom of the figure (asterisk (*) indicates a nonsense mutation). In ATP6V1B2, the R485P mutation is associated with ZLS2, and the R506* mutation is associated with DODD. Schematic diagram was generated using Illustrator for Biological Sequences (IBS) 2.0 [49].

**Figure 5 jdb-13-00004-f005:**
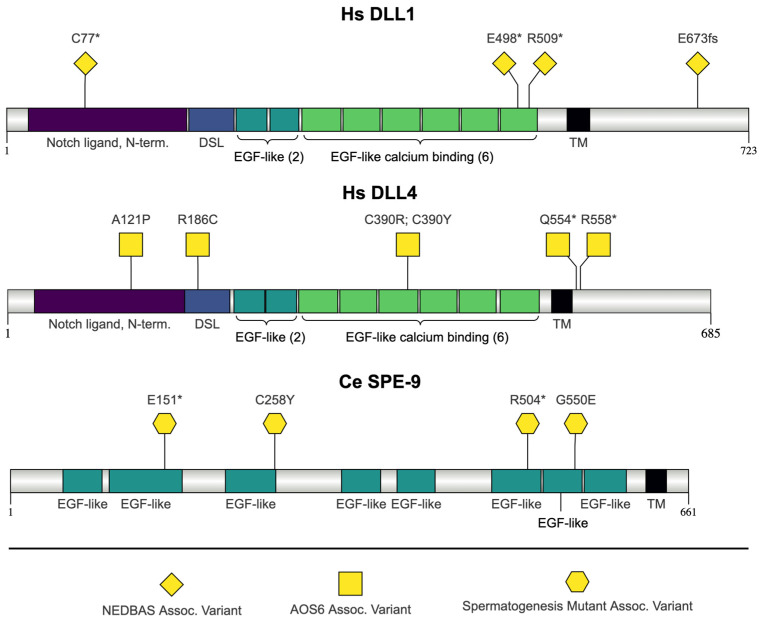
Protein diagram of *C. elegans* SPE-9 and *H. sapiens* DLL1 and DLL4. Diagram depicting full-length protein with protein domains and genetic variants. Human and *C. elegans* genetic variants are labeled with the shapes indicated at the bottom of the figure (asterisk (*) indicates a nonsense mutation). Schematic diagram was generated using Illustrator for Biological Sequences (IBS) 2.0 [49].

**Figure 6 jdb-13-00004-f006:**
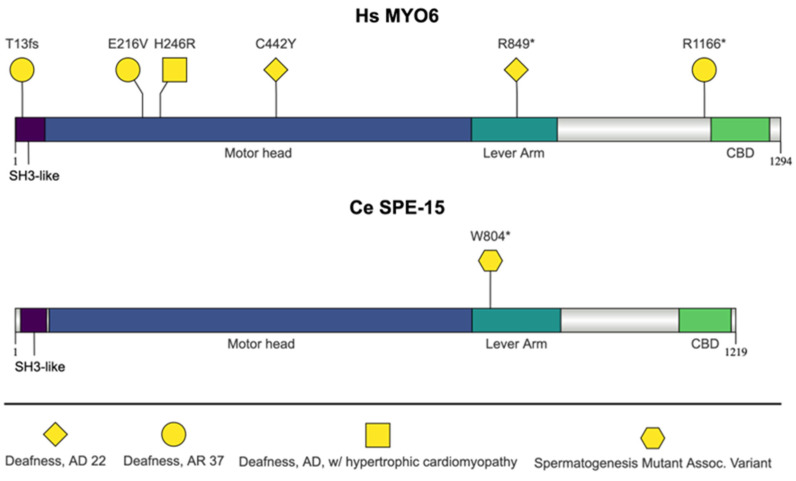
Protein diagram of *C. elegans* SPE-15 and *H. sapiens* MYO6. Diagram depicting full-length protein with protein domains and genetic variants. Human and *C. elegans* genetic variants are labeled with the shapes indicated at the bottom of the figure (asterisk (*) indicates a nonsense mutation). In MYO6, the T14fs, E216V, and R1166* mutations are associated with autosomal recessive (AR) deafness, the C442Y and R849 mutations are associated with autosomal dominant deafness (AD), and the H246R mutation is associated with AD deafness with hypertrophic cardiomyopathy. Schematic diagram was generated using Illustrator for Biological Sequences (IBS) 2.0 [49].

**Figure 7 jdb-13-00004-f007:**
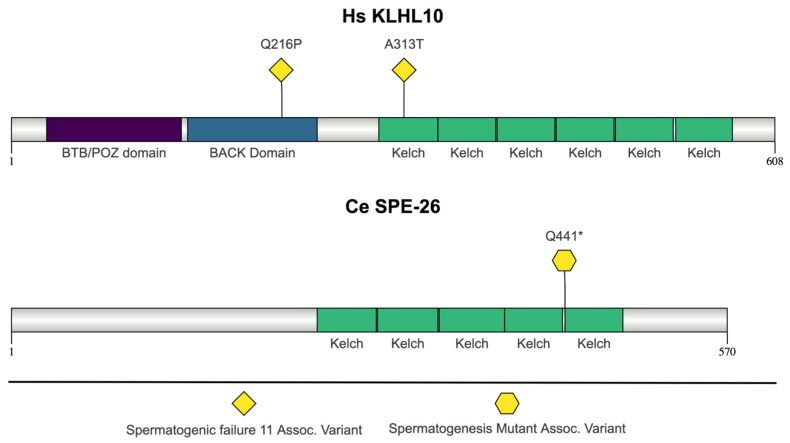
Protein diagram of *C. elegans* SPE-26 and *H. sapiens* KLHL10. Diagram depicting full-length protein with protein domains and genetic variants. Human and *C. elegans* genetic variants are labeled with the shapes indicated at the bottom of the figure (asterisk (*) indicates a nonsense mutation). Schematic diagram was generated using Illustrator for Biological Sequences (IBS) 2.0 [49].

**Figure 8 jdb-13-00004-f008:**
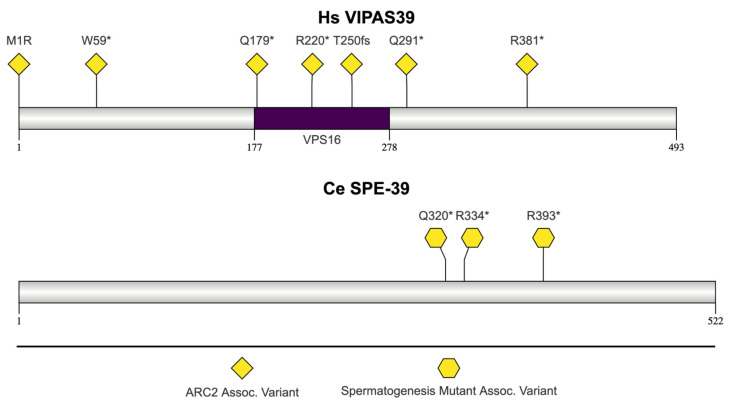
Protein diagram of *C. elegans* SPE-39 and *H. sapiens* VIPAS39. Diagram depicting full-length protein with protein domains and genetic variants. Human and *C. elegans* genetic variants are labeled with the shapes indicated at the bottom of the figure (asterisk (*) indicates a nonsense mutation). Schematic diagram was generated using Illustrator for Biological Sequences (IBS) 2.0 [49].

**Figure 9 jdb-13-00004-f009:**
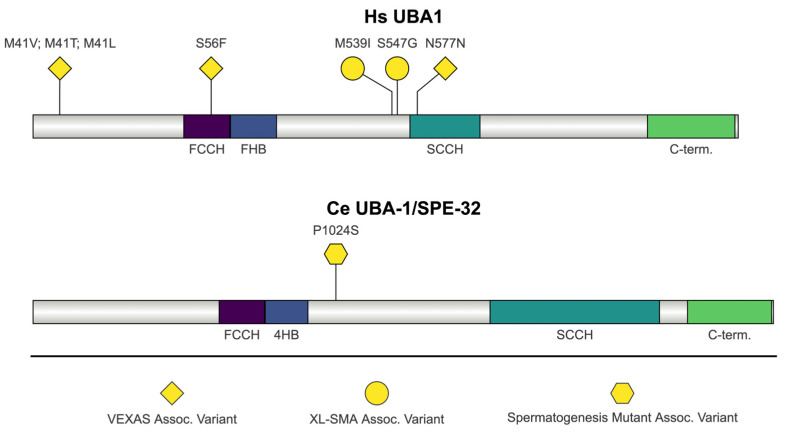
Protein diagram of *C. elegans* UBA-1/SPE-32 and *H. sapiens* UBA1. Diagram depicting full-length protein with protein domains and genetic variants. Human and *C. elegans* genetic variants are labeled with the shapes indicated at the bottom of the figure. The M431V, M41T, M41L, and S56P variants are associated with VEXAS, and the M539I and S547G variants are associated with XL-SMA. Schematic diagram was generated using Illustrator for Biological Sequences (IBS) 2.0 [49].

**Table 1 jdb-13-00004-t001:** Human orthologs of *C. elegans fer* and *spe* genes and OMIM phenotypes.

*C. elegans* genes			Human Orthologs		OMIM Phentoype	
WormBase ID	Common Name	Locus ID	Ensembl ID	HGNC Symbol	Phenotype MIM #	Phenotype
WBGene00001414	*fer-1*	F43G9.6	ENSG00000135636	*DYSF*	253601	Muscular dystrophy, limb-girdle, autosomal recessive 2
					254130	Miyoshi muscular dystrophy 1
					606768	Myopathy, distal, with anterior tibial onset
			ENSG00000088340	*FER1L4*	-	-
			ENSG00000249715	*FER1L5*	-	-
			ENSG00000214814	*FER1L6*	-	-
			ENSG00000138119	*MYOF*	619366	Angioedema, hereditary, 7 ?
			ENSG00000115155	*OTOF*	601071	Auditory neuropathy, autosomal recessive, 1
						Deafness, autosomal recessive 9
WBGene00012933	*mib-1/fer-2/fer-4/spe-16*	Y47D3A.22	ENSG00000101752	*MIB1*	615092	Left ventricular noncompaction 7
			ENSG00000197530	*MIB2*	-	Variant of unknown signficance
WBGene00004959	*spe-5*	Y110A7A.12	ENSG00000116039	*ATP6V1B1*	267300	Distal renal tubular acidosis 2 with progressive sensorineural hearing loss
			ENSG00000147416	*ATP6V1B2*	124480	Deafness, congenital, with onychodystrophy, autosomal dominant
					616455	Zimmermann-Laband syndrome 2
WBGene00004963	*spe-9*	C17D12.6	ENSG00000198719	*DLL1*	618709	Neurodevelopmental disorder with nonspecific brain abnormalities and with or without seizures
			ENSG00000128917	*DLL4*	616589	Adams-Oliver syndrome 6
WBGene00004969	*spe-15*	F47G6.4	ENSG00000196586	*MYO6*	606346	Deafness, autosomal dominant 22
						Deafness, autosomal dominant 22, with hypertrophic cardiomyopathy
					607821	Deafness, autosomal recessive 37
WBGene00004972	*spe-26*	R10H10.2	ENSG00000161594	*KLHL10*	615081	Spermatogenic failure 11
WBGene00004975	*spe-39*	ZC404.3	ENSG00000151445	*VIPAS39*	613404	Arthrogryposis, renal dysfunction, and cholestasis 2
WBGene00006699	*uba-1/spe-32*	C47E12.5	ENSG00000130985	*UBA1*	301054	VEXAS syndrome, somatic
					301830	Spinal muscular atrophy, X-linked 2, infantile
			ENSG00000182179	*UBA7*	-	-

? = the relationship between the phenotype and gene is provisional.

## Data Availability

The original contributions presented in this study are included in the article. Further inquiries can be directed to the corresponding author(s).

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
