# Peer review of "Utilizing C. elegans Spermatogenesis and Fertilization Mutants as a Model for Human Disease"

_jdb, 2025, doi:10.3390/jdb13010004_

Round 1

Reviewer 1 Report

Comments and Suggestions for Authors

This study presents a well-defined review of the utility of Caenorhabditis elegans as a model organism to study genes involved in fertilization, particularly focusing on seven genes that have human orthologs associated with pathogenic phenotypes. The study highlights the critical role C. elegans has played in advancing our understanding of protein domain function and human diseases, specifically through gene orthologs like fer-1 and its human equivalent OTOF (related to hearing loss), spe-39 and VIPAS39 (vacuolar protein sorting), and spe-26 and KLHL10 (spermatogenesis). The review emphasizes the potential of C. elegans to provide insights into the function of understudied genes, which may improve disease diagnosis and clinical decision making.

1, The authors illustrate orthologous gene functions, such as OTOF’s role in hearing loss and VIPAS39’s role in vacuolar protein sorting. However, further mechanistic detail on how C. elegans mutants have specifically contributed to the understanding of these processes would enrich the discussion.

2, While the authors mention the potential of studying understudied and undiscovered genes, the review would benefit from more specific examples of genes that remain unexplored in C. elegans but could provide valuable insights into human disease.

3, The resolution of Table 1 is too low.

Author Response

Thank you very much for taking the time to review this manuscript. The comments and suggestions improved the manuscript greatly. Please find the detailed responses below and the corresponding revisions/corrections highlighted/in track changes in the re-submitted files. We have addressed all of the comments that were made.

Comment 1: The authors illustrate orthologous gene functions, such as OTOF’s role in hearing loss and VIPAS39’s role in vacuolar protein sorting. However, further mechanistic detail on how C. elegans mutants have specifically contributed to the understanding of these processes would enrich the discussion.

Response 1: We added more detail to how FER-1 analysis aided in OTOF function elucidation and treatment and clarified the role of SPE-39 in understanding VIPAS39 function to the discussion (lines 1074-1083).

Comment 2: While the authors mention the potential of studying understudied and undiscovered genes, the review would benefit from more specific examples of genes that remain unexplored in C. elegans but could provide valuable insights into human disease.

Response 2: We added a new paragraph in the Discussion to address this point (lines 1104-1149).

Comment 3: The resolution of Table 1 is too low.

Response 3: We replaced Table 1 with a new edited version. The new file has 600 dpi and was submitted along with the other figures as a .png file.

Reviewer 2 Report

Comments and Suggestions for Authors

Overall, this review will be an important and useful contribution to the field since it brings together material that has not been together in this fashion. It has the potential of serving as a useful jumping off point for individuals who seek to study human disease mutations in a model organism. The final discussion section on phenologs will be helpful for readers who might be mistaken that C. elegans spermatogenesis phenotypes are only relevant to human fertility defects.

The authors should consider including a line drawing of C. elegans spermatogenesis/sperm activation that includes key events and structures (MOs & MO fusion, residual bodies, etc.) discussed throughout the paper (potentially with defects observed in the mutants discussed). Such a figure will make this paper accessible to a broader audience.

Page 2

Line 52 – hermaphrodites produce 300-400 sperm during their final larval stage before switching to oocyte production. Individual males produce sperm continuously, and single males can sire up to a thousand progeny.

Line 59 allows for both the redistribution of proteins necessary for fertilization and the secretion of glycoproteins whose functions remain largely uncharacterized.  Mos are specialized secretory vesicles within spermatids that fused with the…

Line 68  …only a few progenies produced. In Spe hermaphrodites, the majority of oocytes remain unfertilized due to their defective sperm. However, since their oocytes are unaffected, Spe hermaphrodites produce viable progeny when crossed to wild-type males.

Page 4. In the description of Ferlin proteins, please clarify that these proteins lack a transmembrane domain and are instead associated with the membrane by their C2 domains.

Page 5 – 174. Since you mention these other domains, it should be possible to indicate where they are by colored lines below the bar figure.

Page 6

line 185. “altered pseudopods” is vague.  Revise to “fer-1 sperm have abnormally short pseudopods and are non-motile. Notably, their MOs fail to fuse with the plasma membrane.

Line 192 … are nonconditional; sperm with these genetic lesions never fuse their MOs.

Line 217 …that are listed in OMIM.

Page 8 line 328.  Missing period after domain and before the second V1.

Page 9 Line 340 – SPE-5 localizes to the MOs (it isn’t expressed there). Although this section is entitled SPE-5 function in C. elegans, the description of the mutant phenotype is quite minimal. Can it be expanded?

Figure 3 – Please indicate the transmembrane domain.  

Page 12  line 469-472. There are muscle myosins and non-muscle myosins. A subset of non-muscle myosins are called unconventional myosins because they (unlike conventional myosins) move towards the minus end of actin filaments. As written the text implies that unconventional myosins are merely not in muscle. 

Should include Kelleher et al 2000 citation in this section about SPE-15

Page 13 – In the description of SPE-15 the authors need to add/clarify the following points:

a)      Clarify that for many spe-15 alleles, there is a defect in the differential segregation of organelles into the spermatids and away from the residual body (cytoblast). In this case the spermatids individualize but have subsequent defects in sperm activation. Notably when both spe-15 and nmy-2 (non-muscle myosin) are knocked out, residual bodies fail to form altogether. For broader readership, this explanation will need at least a cartoon drawing residual body formation.

b)     Line 482 -  Hu, 2019 paper should be a numbered citation.

c)      Localization of SPE-15 (Hu, 2019) shows that SPE-15 localizes the constriction between the spermatid and residual body – so it is functioning in the separation event as well as during the partitioning that precedes the final scission event.

Page 15 line 553. Again, if there was diagram of the associated events of C. elegans spermatogenesis and the spe-26 defects, the authors could say a bit more about the mutant phenotype. Since the C. elegans protein does not have the BTB and BACK domain, it is worth saying a little bit more about the known diverse functions of the Kelch propeller region – binding and cross-linking actin filaments, serving as an enzyme, binding to other diverse protein partners. We don’t know what spe-26 binds to…

Line 573 – is Klh10 present in the cytoplasm of spermatids or is one of the smaller set of proteins that is actually translated post-meiotically. Please be specific whether it is expressed (transcribed/translated) or present.

Author Response

Thank you very much for taking the time to review this manuscript. The comments and suggestions improved the manuscript greatly. In particular we added a figure (Figure 1) to highlight the events of spermatogenesis and sperm activation that were most relevant for the review.

Please find the detailed responses below and the corresponding revisions/corrections highlighted/in track changes in the re-submitted files. We have addressed all of the comments that were made.

Comment 1: Line 52 – hermaphrodites produce 300-400 sperm during their final larval stage before switching to oocyte production. Individual males produce sperm continuously, and single males can sire up to a thousand progeny.

Response 1: The manuscript was edited to include these sentences (lines 56-58).

Comment 2: Line 59 allows for both the redistribution of proteins necessary for fertilization and the secretion of glycoproteins whose functions remain largely uncharacterized. Mos are specialized secretory vesicles within spermatids that fused with the…

Response 2: The manuscript was edited to reflect this information (lines 63-66).

Comment 3: Line 68 …only a few progenies produced. In Spe hermaphrodites, the majority of oocytes remain unfertilized due to their defective sperm. However, since their oocytes are unaffected, Spe hermaphrodites produce viable progeny when crossed to wild-type males.

Response 3: The manuscript was edited to include these sentences (lines 104-108).

Comment 4: Page 4. In the description of Ferlin proteins, please clarify that these proteins lack a transmembrane domain and are instead associated with the membrane by their C2 domains.

Comment 4: Analysis by Lek (10.1186/1471-2148-10-231) and Dominguez (10.1371/journal.pone.0270188) indicate that the ferlins discussed have a transmembrane domain. In Interpro this is embedded in the C-terminal domain. Figure 2 is consistent with the Interpro representation. We do point out that the C2 domains can bind phospholipids in membranes (lines 218-220).

Comment 6: Page 5 – 174. Since you mention these other domains, it should be possible to indicate where they are by colored lines below the bar figure.

Response 6: We have edited Figure 2 to include those domains and clarified the domain labels and coloring.

Comment 7: line 185. “altered pseudopods” is vague. Revise to “fer-1 sperm have abnormally short pseudopods and are non-motile. Notably, their MOs fail to fuse with the plasma membrane.

Response 7: The manuscript was edited to reflect this information (lines 257-258).

Comment 8: Line 192 … are nonconditional; sperm with these genetic lesions never fuse their MOs.

Response 8: The manuscript was edited to reflect this information (lines 264-265).

Comment 9: Line 217 …that are listed in OMIM.

Response 9: The manuscript was edited to reflect this (lines 318-319).

Comment 10: Page 8 line 328. Missing period after domain and before the second V1.

Response 10: The manuscript was edited (line 549).

Comment 11: Page 9 Line 340 – SPE-5 localizes to the MOs (it isn’t expressed there). Although this section is entitled SPE-5 function in C. elegans, the description of the mutant phenotype is quite minimal. Can it be expanded?

Response 11: The manuscript was edited to clarify the SPE-5 localization (lines 560-562). More detail for SPE-5 function was also added to this section (lines 555-579).

Comment 12: Figure 3 – Please indicate the transmembrane domain.

Response 12: The figure has been edited to reflect this change (it is now Figure 4).

Comment 13: Page 12 line 469-472. There are muscle myosins and non-muscle myosins. A subset of non-muscle myosins are called unconventional myosins because they (unlike conventional myosins) move towards the minus end of actin filaments. As written the text implies that unconventional myosins are merely not in muscle.

Response 13: The section has been edited to reflect this information and for clarity (lines 719-724).

Comment 14: Should include Kelleher et al 2000 citation in this section about SPE-15

Response 14: The citation has been added to this section.

Comment 15: Page 13 – In the description of SPE-15 the authors need to add/clarify the following points:

  1. Clarify that for many spe-15 alleles, there is a defect in the differential segregation of organelles into the spermatids and away from the residual body (cytoblast). In this case the spermatids individualize but have subsequent defects in sperm activation. Notably when both spe-15 and nmy-2 (non-muscle myosin) are knocked out, residual bodies fail to form altogether. For broader readership, this explanation will need at least a cartoon drawing residual body formation.
  2. Line 482 - Hu, 2019 paper should be a numbered citation.
  3. Localization of SPE-15 (Hu, 2019) shows that SPE-15 localizes the constriction between the spermatid and residual body – so it is functioning in the separation event as well as during the partitioning that precedes the final scission event.

Response 15: The manuscript has been edited to reflect these points and for clarification (lines 731-736).

Comment 16: Page 15 line 553. Again, if there was diagram of the associated events of C. elegans spermatogenesis and the spe-26 defects, the authors could say a bit more about the mutant phenotype. Since the C. elegans protein does not have the BTB and BACK domain, it is worth saying a little bit more about the known diverse functions of the Kelch propeller region – binding and cross-linking actin filaments, serving as an enzyme, binding to other diverse protein partners. We don’t know what spe-26 binds to…

Response 16: The manuscript has been edited to provide more detail about the kelch domains (lines 820-822). More detail has been added about the spe-26 mutant phenotype, which is supported by the new Figure 1 (lines 838-845).

Comment 17: Line 573 – is Klh10 present in the cytoplasm of spermatids or is one of the smaller set of proteins that is actually translated post-meiotically. Please be specific whether it is expressed (transcribed/translated) or present.

Response 17: The manuscript has been edited to reflect this (lines 871-874).

Reviewer 3 Report

Comments and Suggestions for Authors

"Utilizing C. elegans Fertilization Mutants as a Model for Human Disease" by Sofia M. Perez, Helena S. Augustineli and Matthew R. Marcello is an interesting perspective on how C. elegans spermatogenesis mutants have and could continue to provide insight into human disease processes affected by homologous genes. However, as presently assembled, the body of data chosen for discussion is incomplete and this should be addressed to make the manuscript a comprehensive compendium of how C. elegans spermatogenesis-defective mutants can play a significant role in analyzing genes affected in human diseases. My recommendations should be straightforward to complete and, if all of the issues discussed below are addressed, this manuscript would be substantially strengthened and become quite suited for publication.

Generally, genes in the spe-9 class are thought to represent the C. elegans "fertilization genes".  This paper discusses one of those genes, spe-9, but not any of the other 9 genes, but it does discuss a few, but not all, of the genes involved in C. elegans spermatogenesis that have human homologs/orthologs implicated in disease processes.  Consequently, the word "fertilization", which has a specific meaning, should be changed to "spermatogenesis" on both the title and throughout the text because this word more accurately represents what is discussed in this manuscript. The egg genes are also mentioned at the beginning, yet not discussed further, so they should either be discussed or removed from the revised manuscript. Additionally, the egg genes distract from the focus on genes involved in spermatogenesis.

The sentence "Some pathogenic variants in OTOF 291can also cause a less frequent temperature‐sensitive version of ANSD [88]." should be explained regarding how temperature sensitivity can occur in a warm-blooded human.

SPE-5: There is no discussion of the fact that SPE-5, which is on chromosome I, has an X chromosome paralog named VHA-12 and that these genes are functionally equivalent but have different expression patterns.  This simplifies explanation of the considerable somatic defects associated with disease shown by loss of function of human vacuolar ATPase.  The reason is the tissue specificity of SPE-5 expression, not because the B subunit is non essential.

"spe‐15 is necessary for the separation of spermatids from residual bodies (RB) (Hu 482

2019). Hu 2019 should be [145] because it is a reference.

do not [17,21,145]. spe‐15(ok153) is a deletion mutation that removes sequences that encode 491

add: "have activation defects" after "not"

26(hc138), is a temperature sensitive mutation thin the splice site in the second intron and 562

change "thin" to "in"

in Figure 6 [166]."Hermaphrodites of all of the alleles are self‐sterile and can produce prog‐" 605

change to: " Hermaphrodites bearing any of these mutant alleles are self‐sterile and can produce prog‐"

"by mutations in VPS33B [27]. Patients with mutations in VIPAS39 display a typical ARC 619"

are you sure [27] is the correct citation here?

The additional genes should have been discussed and they are spe-4 and mib-1, both of which have a Spe phenotype.  Like SPE-5, SPE-4 also has a paralog, which is SEL-12 and the two proteins provide presenilin function in C. elegans; presenilins are implicated in early-onset Alzheimer’s disease.

MIB-1 is the sole C. elegans Mind Bomb ortholog, which is a ubiquitin E3 ligase that is required for processing targets that include Notch (GLP-1 and LIN-12 in C. elegans. Strangely, mutation is C. elegans mib-1 are all temperature sensitive Spe, despite its expression and roles in somatic tissues; this expression in both the male testis and somatic tissues is also seen with spe-5 and spe-39. Mutations in Mind Bomb are implicated in a variety of human diseases including congenital heart disease, bicuspid aortic valve and left ventricular noncompaction and in the diagnosis of gliomas and follicular lymphomas. Many of the phenotypic consequences of UBA-1, which is a ubiquitin E1 ligase (and the only one found in C. elegans) that is discussed in this manuscript, could be mediated through MIB-1 activity.

The Discussion section is quite brief and does not comprehensively discuss the body of data presented in this manuscript. This situation should be addressed.

Author Response

Thank you very much for taking the time to review this manuscript. The comments and suggestions improved the manuscript greatly. We have made significant changes, including adding a section (MIB-1), expanding the Discussion, and reframing the review more around spermatogenesis.

Please find the detailed responses below and the corresponding revisions/corrections highlighted/in track changes in the re-submitted files. We have addressed all of the comments that were made.

Comment 1: Generally, genes in the spe-9 class are thought to represent the C. elegans "fertilization genes". This paper discusses one of those genes, spe-9, but not any of the other 9 genes, but it does discuss a few, but not all, of the genes involved in C. elegans spermatogenesis that have human homologs/orthologs implicated in disease processes. Consequently, the word "fertilization", which has a specific meaning, should be changed to "spermatogenesis" on both the title and throughout the text because this word more accurately represents what is discussed in this manuscript. The egg genes are also mentioned at the beginning, yet not discussed further, so they should either be discussed or removed from the revised manuscript. Additionally, the egg genes distract from the focus on genes involved in spermatogenesis.

Response 1:  We have made significant revisions in response to your review, including adding a section about MIB1 function and framing the review more around spermatogenesis. We also moved any discussion of the egg genes to the Discussion section. We chose frame the review around both spermatogenesis and fertilization. The main reason is we believe that using the term "fertilization mutants" in addition to spermatogenesis mutants will signal to readers that there are mutants that make sperm that are morphologically normal but cannot fuse with the egg. Our belief is readers from other fields would believe that spermatogenesis has occured normally in these mutants.  Furthermore, the historical and now defunct conventions for naming are memorialized in the gene names and keeping "fertilization" as part of the framing will help readers understand the connection between fer-1 in particular and the protein functions discussed.

Comment 2: The sentence "Some pathogenic variants in OTOF 291can also cause a less frequent temperature‐sensitive version of ANSD [88]." should be explained regarding how temperature sensitivity can occur in a warm-blooded human.

Response 2: The manuscript has been edited to clarify this point (lines 396-400).

Comment 3: SPE-5: There is no discussion of the fact that SPE-5, which is on chromosome I, has an X chromosome paralog named VHA-12 and that these genes are functionally equivalent but have different expression patterns. This simplifies explanation of the considerable somatic defects associated with disease shown by loss of function of human vacuolar ATPase. The reason is the tissue specificity of SPE-5 expression, not because the B subunit is non essential.

Response 3: The manuscript has been edited to reflect this (lines 562-565).

Comment 4: "spe‐15 is necessary for the separation of spermatids from residual bodies (RB) (Hu 2019). Hu 2019 should be [145] because it is a reference.

Response 4: The manuscript has been edited to reflect this (line 729).

Comment 5: do not [17,21,145]. spe‐15(ok153) is a deletion mutation that removes sequences that encode 491 add: "have activation defects" after "not"

Response 5: The manuscript has been edited to reflect this (lines 760-761).

Comment 6: 26(hc138), is a temperature sensitive mutation thin the splice site in the second intron and 562 change "thin" to "in"

Response 5: The manuscript has been edited to reflect this (line 856).

Comment 6: in Figure 6 [166]."Hermaphrodites of all of the alleles are self‐sterile and can produce prog‐" 605 change to: " Hermaphrodites bearing any of these mutant alleles are self‐sterile and can produce prog‐"

Response 6: The manuscript has been edited to reflect this (line 911).

Comment 7: "by mutations in VPS33B [27]. Patients with mutations in VIPAS39 display a typical ARC 619"

are you sure [27] is the correct citation here?

Response 7: the citations have been checked and updated (line 931).

Comment 8: The additional genes should have been discussed and they are spe-4 and mib-1, both of which have a Spe phenotype. Like SPE-5, SPE-4 also has a paralog, which is SEL-12 and the two proteins provide presenilin function in C. elegans; presenilins are implicated in early-onset Alzheimer’s disease.

MIB-1 is the sole C. elegans Mind Bomb ortholog, which is a ubiquitin E3 ligase that is required for processing targets that include Notch (GLP-1 and LIN-12 in C. elegans. Strangely, mutation is C. elegans mib-1 are all temperature sensitive Spe, despite its expression and roles in somatic tissues; this expression in both the male testis and somatic tissues is also seen with spe-5 and spe-39. Mutations in Mind Bomb are implicated in a variety of human diseases including congenital heart disease, bicuspid aortic valve and left ventricular noncompaction and in the diagnosis of gliomas and follicular lymphomas. Many of the phenotypic consequences of UBA-1, which is a ubiquitin E1 ligase (and the only one found in C. elegans) that is discussed in this manuscript, could be mediated through MIB-1 activity.

Response 8: SPE-4 does not have a ortholog identified in the Alliance database, which was a requirement for inclusion in the review. MIB-1 does have an ortholog with an associated phenotype and we have added a section (Section 2.2) and figure 3 (Figure 3) to discuss the function of MIB-1 and its orthologs (lines 445-539).

Comment 9: The Discussion section is quite brief and does not comprehensively discuss the body of data presented in this manuscript. This situation should be addressed.

Response 9: We have expanded the Discussion to include more detail about fer-1 and spe-39, information about C. elegans phenologs, and a new paragraph about how new genes will fall under the scope of the review as we understand more about human and C. elegans biology.

Reviewer 4 Report

Comments and Suggestions for Authors

Perez et. al., review several C. elegans genes required for spermatogenesis, sperm function, and fertilization along with their corresponding human orthologs. This review carefully discusses seven C. elegans genes with human orthologs with associated disease phenotypes. The authors present schematic diagrams of the C. elegans genes, their human orthologs, and the location and amino acid changes of associated genetic variants. The strengths of this review are the brief overview of the known function of the C. elegans protein along with the following comprehensive summary of human disease variants and the known molecular consequences of the variants. Another strength is the brief discussion on the significance and relevance the study of C. elegans fertilization and spermatogenesis mutants have to human disease and potential treatments.

Comments:

1) A schematic legend for the figures to indicate the domains (by color), rather than writing it next to it on the schematic, along with the diamond for the variants would be very helpful.  This is especially important for the proteins with long domain names (e.g., Figure 2: ATPase, F1/V1/A1 complex, alpha/beta subunit, nucleotide-binding domain). The other figures that would benefit from this are Figures 4 and 7.

2) The Figure 1 legend lists the variants of DYSF, MYOF, and OTOF that are associated with specific diseases. This should be included for the other figure legends that have more than one disease associated with a particular gene (Figure 2 and Figure 3).

3) Discussion: I think it would make more sense to move the paragraph on phenologs earlier in the discussion. Maybe after the first paragraph.

4) Section 2.3.1: A brief sentence describing LAG-2 is needed (ligand for a C. elegans Notch receptor).

5) Page 8 description of the DFBN9: Is the Glu747Ter more severe than the truncated Tyr730Ter?

Please explain why the pArg1792His is likely pathologic.

Is “pathogenic” (page 8 lines 299 and 314) correct? I believe it should be pathologic.

Line 304: Do the mice have phenotypes? Otherwise, why is it considered to be pathologic?

Last sentence of this section (line 318): What is meant by the p.Arg1939Gln mutation is predicted to be damaging? What is being damaged?

5) Page 7: The paragraph that consists of lines 254-255 should be moved to after the paragraph on the mutations associated with both MMD and LGMD2B. It could also be combined with the final paragraph (mutations associated with all 3 diseases).

6) Section 2.6.2: Line 605-606 says that spe-39 mutant hermaphrodites can produce progeny after mating to males. Is this only with wild-type males? Do spe-39 mutant males produce cross progeny?

FB needs to be defined.

The last line in this paragraph describes an RNAi experiment. Please clarify if these experiments were done in C. elegansor in human cultured cells as both were used in this paper.

7) Page 18 line 688 there is a discrepancy in the amino acid of the missense allele [“resulting in a serine to proline substitution (p.Ser56Phe)”]. Is it proline or phenylalanine? Figure 7 shows it as proline.

Minor modifications:

Page 2 line 85: Add the word genes after “The human orthologs to C. elegans eggfer, and spe.”

Page 2 line 61: remove “are” from MOs are specialized secretory vesicles are contained…”

Page 3 title to Table 1 change “mutants” to “genes.”

Page 6 line 209: change “an” to “of.”

Page 6 line 227: Change “and” in “mutation is the result of a transition mutation that activates and” to “an.”

Page 8 line 296: Remove “to” in “a truncated protein to that leads to a severe…”

Page 10 line 374: Change “highly conserved mutation” to highly conserved gene.

Page 11 line 425: Change “eight” to “eighth”

Page 12 line 459: Change “contain” to “containing”

Page 12 line 475: Add a hyphen to coiled-coil

Page 13 lines 495-496: I think a word needs to be removed from “adaptor proteins to regulated coordinate the…”

Page 15 line 562: Change “thin” to “within”

Page 16 line 617: Change “hare” to “are”

Page 19 line 731: Remove “are” from “necessary for fertilization are represent phenologs…”

Author Response

Thank you very much for taking the time to review this manuscript. The comments and suggestions improved the manuscript greatly. In particular, we appreciate the suggestions to clarify the figures.

Please find the detailed responses below and the corresponding revisions/corrections highlighted/in track changes in the re-submitted files. We have addressed all of the comments that were made.

Comment 1: A schematic legend for the figures to indicate the domains (by color), rather than writing it next to it on the schematic, along with the diamond for the variants would be very helpful.  This is especially important for the proteins with long domain names (e.g., Figure 2: ATPase, F1/V1/A1 complex, alpha/beta subunit, nucleotide-binding domain). The other figures that would benefit from this are Figures 4 and 7.

Response 1: We added a figure (Figure 1) to highlight the events of spermatogenesis and sperm activation that were most relevant for the review. We have edited all figures to clarify the domain labels and coloring, which are designed to be distinguishable even in black and white).

Comment 2: The Figure 1 legend lists the variants of DYSF, MYOF, and OTOF that are associated with specific diseases. This should be included for the other figure legends that have more than one disease associated with a particular gene (Figure 2 and Figure 3).

Response 2: We have added different shapes to indicate the types of variants to all figures.

Comment 3: Discussion: I think it would make more sense to move the paragraph on phenologs earlier in the discussion. Maybe after the first paragraph.

Response 3: The paragraph has been moved to the suggested location

Comment 4: Section 2.3.1: A brief sentence describing LAG-2 is needed (ligand for a C. elegans Notch receptor).

Response 4: A sentence has been added to reflect this (lines 641-644).

Comment 5: 

  • Page 8 description of the DFBN9: Is the Glu747Ter more severe than the truncated Tyr730Ter?
  • Please explain why the pArg1792His is likely pathologic.
  • Is “pathogenic” (page 8 lines 299 and 314) correct? I believe it should be pathologic.
  • Line 304: Do the mice have phenotypes? Otherwise, why is it considered to be pathologic?
  • Last sentence of this section (line 318): What is meant by the p.Arg1939Gln mutation is predicted to be damaging? What is being damaged?

Response 5:

  • The manuscript was edited to reflect the fact that these two truncated proteins have the same diagnosis (line 402-405).
  • The manuscript was edited to explain the rationale for the pathogenicity of p.Arg1792His (lines 407-408).
  • Pathogenic is the correct term that is used widely. Pathogenic generally refers to the condition. We kept the word pathologic in line 428 but used quotations to reflect that this is the word that is used in the reference.
  • More detail about the impact of p.Pro50Arg has been added (line 429).
  • More detail about the impact of p.Arg1939Gln has been added (line 443).

Comment 6: Page 7: The paragraph that consists of lines 254-255 should be moved to after the paragraph on the mutations associated with both MMD and LGMD2B. It could also be combined with the final paragraph (mutations associated with all 3 diseases).

Response 6: The manuscript has been edited to reflect this suggestion (lines 368-373).

Comment 7:

  • Section 2.6.2: Line 605-606 says that spe-39 mutant hermaphrodites can produce progeny after mating to males. Is this only with wild-type males? Do spe-39 mutant males produce cross progeny?
  • FB needs to be defined.
  • The last line in this paragraph describes an RNAi experiment. Please clarify if these experiments were done in C. elegansor in human cultured cells as both were used in this paper.

 Response 7:

  • The manuscript has been edited to clarify this (lines 911-912).
  • The FB is now defined in Figure 1.
  • The manuscript has been edited to clarify this (lines 916-918).

Comment 8: Page 18 line 688 there is a discrepancy in the amino acid of the missense allele [“resulting in a serine to proline substitution (p.Ser56Phe)”]. Is it proline or phenylalanine? Figure 7 shows it as proline.

Response 8: The correct amino acid is Phe. The figure has been edited (now Figure 9) to reflect this.

All minor modifications were made, except minor modification #8. Please see track changes version for more detail.

Minor modifications:

  1. Page 2 line 85: Add the word genes after “The human orthologs to C. elegans eggfer, and spe.”
  2. Page 2 line 61: remove “are” from MOs are specialized secretory vesicles are contained…”
  3. Page 3 title to Table 1 change “mutants” to “genes.”
  4. Page 6 line 209: change “an” to “of.”
  5. Page 6 line 227: Change “and” in “mutation is the result of a transition mutation that activates and” to “an.”
  6. Page 8 line 296: Remove “to” in “a truncated protein to that leads to a severe…”
  7. Page 10 line 374: Change “highly conserved mutation” to highly conserved gene.
  8. Page 11 line 425: Change “eight” to “eighth”
  9. Page 12 line 459: Change “contain” to “containing”
  10. Page 12 line 475: Add a hyphen to coiled-coil
  11. Page 13 lines 495-496: I think a word needs to be removed from “adaptor proteins to regulated coordinate the…”
  12. Page 15 line 562: Change “thin” to “within”
  13. Page 16 line 617: Change “hare” to “are”
  14. Page 19 line 731: Remove “are” from “necessary for fertilization are represent phenologs…”

Round 2

Reviewer 3 Report

Comments and Suggestions for Authors

 Utilizing C. elegans Spermatogenesis and Fertilization Mutants 2 as a Model for Human Disease  by Sofia M. Perez, Helena S. Augustineli and Matthew R. Marcello is a well-written revision of their article that is now ready for publication, subject to some very minor grammatical issues shown below:

Line 73: Change: "unwanted" to "no longer needed"

Line 332: Change: "support this [105]" to "support this contention [105]"

Line 762: Change: “are predicted to resulting in” to “are predicted to result in”

Line 855: Change: “foundational how” to “foundational for understanding how”

Line 862: Change: “spermatogenesis fertilization” to “spermatogenesis and/or fertilization”

Line 881: Change: “with defined human orthologs with phenotypes cataloged” to “with defined human orthologs that have phenotypes cataloged

Lines 889-891: Change: “Additionally, more C. elegans genes that are important for spermatogenesis and fertilization being discovered, including two new sperm genes necessary for fertilization, spe-36 and spe-51” toAdditionally, C. elegans genes that are important for spermatogenesis and fertilization continue to be discovered, including two new sperm-expressed genes necessary for fertilization, spe-36 and spe-51.

Author Response

Comments:

Utilizing C. elegans Spermatogenesis and Fertilization Mutants 2 as a Model for Human
Disease by Sofia M. Perez, Helena S. Augustineli and Matthew R. Marcello is a well-written
revision of their article that is now ready for publication, subject to some very minor
grammatical issues shown below:
Line 73: Change: "unwanted" to "no longer needed"
Line 332: Change: "support this [105]" to "support this contention [105]"
Line 762: Change: “are predicted to resulting in” to “are predicted to result in”
Line 855: Change: “foundational how” to “foundational for understanding how”
Line 862: Change: “spermatogenesis fertilization” to “spermatogenesis and/or fertilization”
Line 881: Change: “with defined human orthologs with phenotypes cataloged” to “with defined
human orthologs that have phenotypes cataloged
Lines 889-891: Change: “Additionally, more C. elegans genes that are important for
spermatogenesis and fertilization being discovered, including two new sperm genes necessary for
fertilization, spe-36 and spe-51” to “Additionally, C. elegans genes that are important for
spermatogenesis and fertilization continue to be discovered, including two new sperm-expressed
genes necessary for fertilization, spe-36 and spe-51.

Response: We have made all of the suggested changes outlined by the reviewer.  The seven changes are in lines 76, 430, 933, 1040, 1047, 1067, and 1076 in the "jdb-3277148_revised_trackchanges" version of the manuscript.